# A transcriptional response to replication stress selectively expands a subset of *Brca2*-mutant mammary epithelial cells

Maryam Ghaderi Najafabadi[1], G. Kenneth Gray [2], Li Ren Kong [3,4,5,6], Komal Gupta[3,4,6], David Perera[3], Huw Naylor[7], Joan S. Brugge [2], Ashok R. Venkitaraman [3,4,8] ✉ & Mona Shehata [1,3] ✉

Germline *BRCA2* mutation carriers frequently develop luminal-like breast cancers, but it remains unclear how *BRCA2* mutations affect mammary epithelial subpopulations. Here, we report that monoallelic $Brca2^{mut/WT}$ mammary organoids subjected to replication stress activate a transcriptional response that selectively expands Brca2$^{mut/WT}$ luminal cells lacking hormone receptor expression (HR-). While CyTOF analyses reveal comparable epithelial compositions among wildtype and Brca2$^{mut/WT}$ mammary glands, Brca2$^{mut/WT}$ HR- luminal cells exhibit greater organoid formation and preferentially survive and expand under replication stress. ScRNA-seq analysis corroborates the expansion of HR- luminal cells which express elevated transcript levels of Tetraspanin-8 (*Tspan8*) and *Thrsp*, plus pathways implicated in replication stress survival including Type I interferon responses. Notably, CRISPR/Cas9-mediated deletion of *Tspan8* or *Thrsp* prevents Brca2$^{mut/WT}$ HR- luminal cell expansion. Our findings indicate that Brca2$^{mut/WT}$ cells activate a transcriptional response after replication stress that preferentially favours outgrowth of HR- luminal cells through the expression of interferon-responsive and mammary alveolar genes.

Women who inherit pathogenic germline *BRCA2* mutations exhibit an increased risk of developing luminal-subtype breast cancers[1,2]. What drives the development of these malignancies remains unclear. There is compelling evidence that BRCA2 is an essential component of the cellular response to DNA replication stress[3,4], which increases the frequency of stalled DNA replication forks. Cells lacking Brca2 exhibit chromosomal instability during cell division[5,6] accompanied by defective DNA repair by homologous DNA recombination[7]. Whereas mammary myoepithelial and luminal cells lacking Brca2 exhibit defects in DNA repair by homologous recombination[8], cells bearing monoallelic *BRCA2* mutations retain these functions[5,6,8]. How mammary cells in

*BRCA2* mutation carriers respond to replication stress is unclear. As all mammary epithelial lineages have proliferative ability[9-11], and therefore experience replication stress. Yet, clinical evidence demonstrates that *BRCA2* carriers tend to develop luminal-like breast cancers[1,2]. These observations raise the possibility that monoallelic *Brca2* mutations may differentially affect replication stress responses in different mammary epithelial subtypes.

An alternative or complementary possible mechanism by which inherited *BRCA2* mutations may enhance breast cancer risk is by causing defects in the differentiation or outgrowth of selected mammary epithelial subpopulations. This possibility has been best

[1]Department of Oncology, University of Cambridge, Cambridge, UK. [2]Department of Cell Biology, Harvard Medical School (HMS), Boston, MA, USA. [3]MRC Cancer Unit, University of Cambridge, Cambridge, UK. [4]The Cancer Science Institute of Singapore, National University of Singapore, Singapore, Singapore. [5]Department of Pharmacology, NUS School of Medicine, National University of Singapore, Singapore, Singapore. [6]NUS Centre for Cancer Research, National University of Singapore, Singapore, Singapore. [7]Cancer Research UK Cambridge Institute, University of Cambridge, Cambridge, UK. [8]Institute of Molecular & Cellular Biology Agency for Science, Technology and Research (A∗STAR), Singapore, Singapore. ✉e-mail: arv22@nus.edu.sg; ms2140@cam.ac.uk

studied in germline BRCA1-associated tumours, a related tumour suppressor gene whose inactivation predisposes women to triple-negative breast cancer subtypes. Germline *BRCA1* mutation carriers exhibit expanded mammary luminal progenitor populations compared to non-carrier individuals prior to cancer onset[12,13], and *BRCA1*-mutant breast tumours may thus originate from this specific cell-of-origin[12,14]. However, there is relatively little information concerning comparable differences in the breasts of *BRCA2* mutation carriers. *BRCA2* mutation carriers develop functional breast tissues containing all epithelial subpopulations[13,15,16], but reported differences in epithelial dynamics have differed among studies of *BRCA2* carriers[15,17–20]. For example, myoepithelial cells from *BRCA2* mutation carriers have relatively minor molecular differences compared to *BRCA1* mutation carriers[21], whereas there are conflicting reports concerning more substantive differences regarding the progenitor capacity of luminal populations in *BRCA2* mutation carriers[13,15,16].

A primary barrier to the assessment of the effects of *BRCA2* heterozygosity on the breast is the inherent clinical variability of the samples studied. Studies of human tissue are often relatively small and may be confounded by other variables affecting breast biology, such as age, parity, and hormonal status. Additionally, multitudinous pathogenic *BRCA2* mutations have been reported in the Clinvar database (https://www.ncbi.nlm.nih.gov/clinvar/), and each may have a distinct effect on mammary homoeostasis. To permit more controlled investigations, we used a genetically engineered murine model carrying a monoallelic germline *Brca2* truncating mutation[22], which recapitulates pathogenic human variants in the so-called breast and ovarian cluster region of BRCA2. Across reproductive stages, this and similar murine models display ostensibly histologically normal mammary glands[8,22,23], suggesting that mammary lineage trajectory is unperturbed by *Brca2* heterozygosity or haploinsufficiency. However, this murine model has not been evaluated using more comprehensive single-cell technologies, nor have the effects of age or hormonal status been assessed in it.

By recapitulating many aspects of in vivo physiology and behaviours, mammary organoids have emerged as a promising in vitro model to enable complex examinations of tissue physiology, particularly over prolonged timeframes, which were hitherto infeasible. Organoids permit well-controlled perturbation experiments, with reduced variability and higher throughput, besides reduced time, cost, and animal usage, compared to similar in vivo experiments. Furthermore, organoids are amenable to genetic manipulation, thus permitting more dynamic and complete genetic modelling of normal tissues.

In this study, we investigated the effect of replication stress on mammary organoids derived from transgenic strains bearing a monoallelic truncating Brca2 mutation (Brca2^mut/WT) compared with wild-type controls (wild type). Histology and CyTOF analyses showed a similar proportional representation of epithelial subtypes in wild-type and Brca2^mut/WT mammary glands. Mammary organoid assays revealed organoid formation capacity was increased in Brca2^mut/WT HR− luminal cell subpopulations. Single exposure of hydroxyurea (HU) to induce replication stress showed wild-type and Brca2^mut/WT mammary cells responded in equivalent manners. However, prolonged HU exposure caused a selective expansion of the Brca2^mut/WT HR− luminal compartment. Single-cell RNA sequencing (scRNA-seq) analyses revealed upregulation of interferon response genes and an increase in the HR- luminal cycling cell cluster proportion. This cell cluster contained transcripts of *Tspan8* and *Thrsp*, both of which are expressed in mammary luminal cells[20,24]. Exposure to inhibitors, or deletion of the HR- luminal cycling cluster via CRISPR/Cas9-mediated Tspan8 and Thrsp knockout demonstrated that Brca2 heterozygous cells activate these pathways to circumvent replication stress-induced cell death and promote cell proliferation and survival.

## Results

### Brca2^mut/WT mammary HR− luminal cells have increased organoid formation capacity

We collected mammary glands from three young (3 months) and four aged (7–10 months) *Brca2*^mut/WT or wild-type mice. Histological sections showed similar morphological appearance (Fig. 1A) and the presence of myoepithelial and luminal lineages (Fig. 1B and Supplementary Fig. 1a). To examine the epithelial proportions, we dissociated mammary glands from 3 young and 5–6 aged wild-type or Brca2^mut/WT mice and analysed the differential expression of CD49f and EpCAM using flow cytometry. The basal and stromal subpopulations contained comparable cell numbers between both genotypes in all ages (Fig. 1C). The luminal population was further fractionated using expression of CD49b and Sca1 to distinguish HR+ and HR- luminal cells. No observable cell proportion differences were detected (Fig. 1C).

As the mammary gland is highly heterogenous with many complex cell states, we generated a mammary CyTOF mass cytometry panel consisting of 39-metal tagged antibodies, which recognise proteins associated with known mammary cell functions. CyTOF was carried out on oestrous-matched young or aged wild-type and Brca2^mut/WT mammary glands. Lineage-relevant markers (CD31, CD45 and Ter119) were used to exclude haemopoietic, endothelial and stromal cells from the analysis. Based on the expression levels of the 39 markers, three clusters were evident, representing the three main epithelial lineages (Fig. 1D). Basal and HR+ luminal subpopulations displayed no significant differences in protein expression irrespective of genotype or age. Several HR− luminal markers, including CD61, SSEA-4, E-Cadherin and CD14 displayed slightly increased expression in the Brca2^mut/WT aged HR− luminal populations compared to wild-type counterparts (Fig. 1D); however, these were not statistically significant. Protein expression of CD14 and E-Cadherin on tissue sections further confirmed these results (Fig. 1E and Supplementary Fig. 1b, c). Thus, our findings suggest that loss of a single Brca2 allele in an unchallenged (i.e., absent additional known tumour driver mutations or genotoxic exposures) mammary gland does not affect the generation or maintenance of epithelial lineages. In addition, we did not detect significant differences in epithelial cell proportion or differentiation state based on age. We therefore elected to utilise aged (9–11 months) mammary tissue for further experiments, because this age in mice corresponds to perimenopausal middle age in humans[25], at which time the incidence of breast cancer in female BRCA2 mutation carriers increases.

Three-dimensional (3D) organoids cultured from mammary tissues display a close resemblance to the epithelial composition of mammary glands, displaying budding structures that mimic branching morphogenesis[26]. A recently developed culture medium for mouse mammary organoid propagation has allowed organoids to be generated from a single cell. By seeding single mammary cell suspensions in Matrigel (Supplementary Fig. 1d), we established a diverse range of organoids composed of spherical-like and multi-lobed structures (Supplementary Fig. 1e). Organoids started to form by day 5/6 and developed within 7–12 days (Supplementary Fig. 1e). Whole-mount staining confirmed the existence of multilineage organoids composed of a myoepithelial/basal outer layer (K14 expression) and an inner E-Cadherin-positive luminal layer containing both Progesterone receptor (PR)-positive and -negative cells (Supplementary Fig. 1f). Collectively, these findings demonstrate that a single mammary cell-derived organoid recapitulates in vivo mammary gland features.

Next, we tested whether Brca2^mut/WT mammary epithelial cells have differential organoid formation capacity. Basal, HR- luminal, and HR+ luminal populations were flow-sorted from four aged (8–11 months) mice and seeded into 3D cultures. All three populations were able to form 3D structures (Supplementary Fig. 1g). The basal and HR- luminal populations generated both spheroid-like and multi-

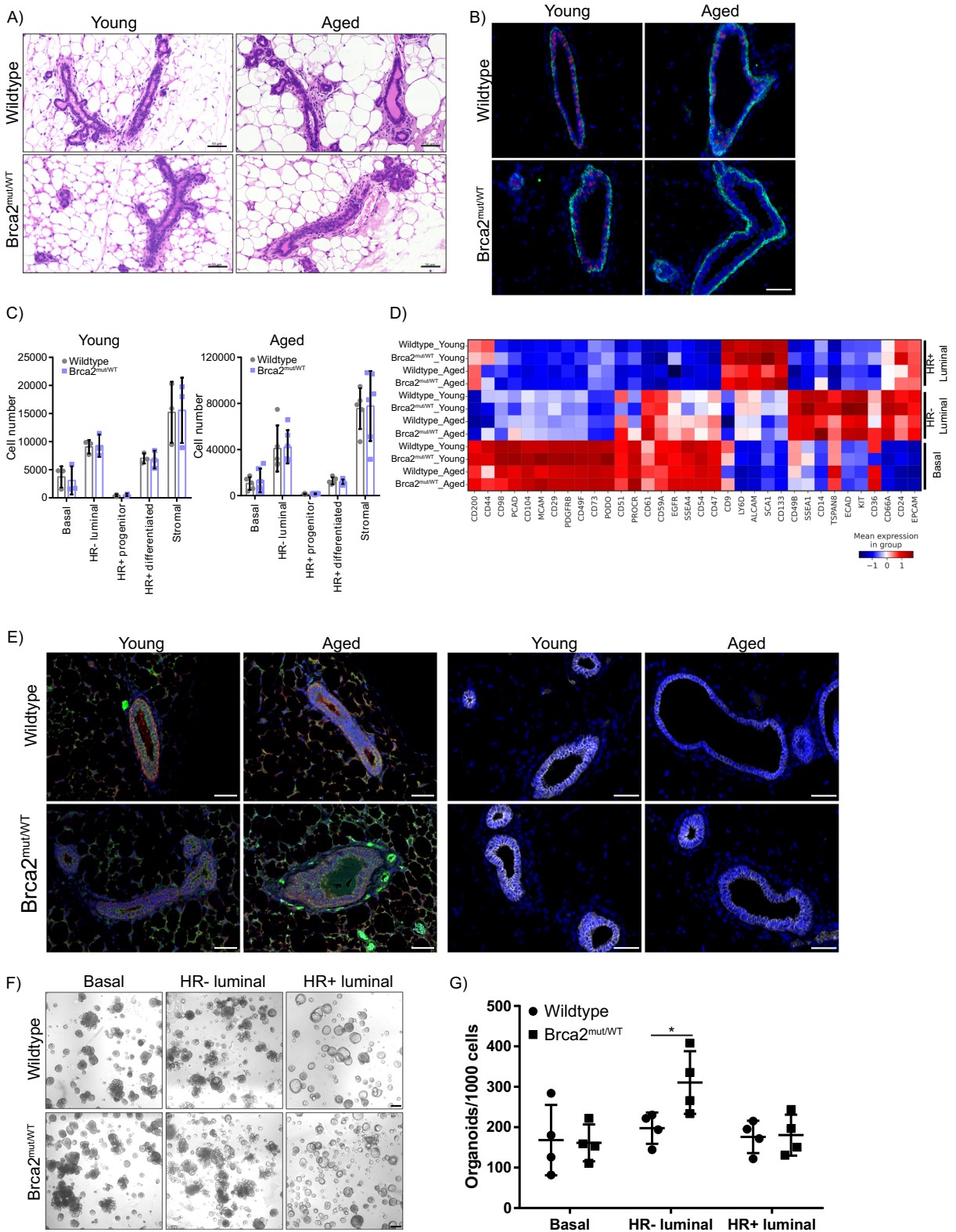

lobed structures, indicating strong organoid formation ability (Fig. 1F and Supplementary Fig. 1g). Whole-mount staining confirmed the presence of both single lineage and multilineage organoids (Supplementary Fig. 1h), confirming the capacity of basal and HR− luminal cells to generate multiple epithelial lineages in vitro, consistent with other reports[27,28]. The HR+ luminal population generated mainly spheroid-like structures in which most organoids were PR+, indicating a lineage-restricted cell type (Supplementary Fig. 1h),

consistent with previous in vitro and in vivo studies[27,29,30]. While basal and HR+ luminal cells generated similar numbers of organoids irrespective of genotype (Fig. 1G), Brca2[mut/WT] HR− luminal cells displayed significantly increased organoid formation capacity (Fig. 1G). In summary, we show that Brca2[mut/WT] HR− luminal cells, while relatively equal in cell proportion, have greater proliferative capacity than their wild-type counterparts when propagated from single cells into organoids.

**Fig. 1 | Brca2$^{mut/WT}$ mammary glands are morphologically similar to wild-type mammary glands in young and aged mice. A** H&E staining of paraffin-embedded sections from young adult (3 months) or aged (8+ months) wild-type or Brca2$^{mut/WT}$ mammary tissues. Images are representative of 3–4 mice/group. Scale bars, 50 μm. **B** PR (magenta) and K14 (green) in young and aged wild-type or Brca2$^{mut/WT}$ mammary glands. Images representative of 3–4 mice/group. Tissues were counter-labelled with 4′,6-diamidino-2-phenylindole (DAPI, blue). Scale bars, 50 μm. **C** Absolute cell number of different mammary epithelial cell subpopulations of young or aged wild-type or Brca2$^{mut/WT}$ mice as determined by flow cytometry. Mean +/− SD of $n = 3$ young independent mice per group and $n = 5$ wild-type or $n = 6$ Brca2$^{mut/WT}$ aged independent mice per group. A two-way ANOVA followed by Tukey's multiple comparison test showed no significant differences. Source data are provided. **D** Heatmap showing expression levels of the indicated markers ($y$ axis) in the epithelial subpopulations from young and aged wild-type or Brca2$^{mut/WT}$ mammary glands (red = high expression, blue = low expression). Data depicts the mean expression of three independent mice per group. **E** Immunostaining of CD14 (green) and CD36 (magenta) (left) and E-Cadherin (grey, right) and DAPI (Blue) in young and aged wild-type and Brca2$^{mut/WT}$ mammary glands. Images representative of 3–4 individual mice/group. Scale bars, 50 μm. **F** Representative brightfield images of wild-type or Brca2$^{mut/WT}$ organoids derived from basal, HR− or HR+ luminal cells. Scale bar, 200 μm. **G** Quantification of the number of wild-type or Brca2$^{mut/WT}$ organoids formed from the different epithelial populations. Data presented as the mean +/− SD (organoids derived from $n = 4$ independent mice). Mann–Whitney two-tailed unpaired test was performed, *$P = 0.04$. Source data are provided as a Source Data file.

## Effect of transient replicative stress on wild-type and Brca2$^{mut/WT}$ mammary organoids

To assess the effects of replication stress on the epithelial subpopulations from both genotypes, we established organoids from single mammary cells derived from five to six mice (9–11 months). On day 7, organoids were exposed to low (1 mM) (Fig. 2) or high (4 mM) (Supplementary Fig. 2) hydroxyurea (HU, which stalls DNA replication) and stained for γH2AX as a reporter of double-strand DNA breaks (Fig. 2A). As expected, γH2AX-positive cells, measured by flow cytometry, were increased in the HU-treated organoids. However, no differences were observed in γH2AX+ cells detected between the genotypes (Fig. 2B and Supplementary Fig. 2a), nor were there any changes in the proportion of the epithelial cell types after HU exposure (Fig. 2C and Supplementary Fig. 2b). These observations suggest that mammary epithelial cells carrying monoallelic Brca2 mutations retain a normal capacity to respond acutely to replication stress.

To examine the acute effects of replication stress on cell division, we administered BrdU to label the DNA of cells in the S phase of the cell cycle and assessed BrdU positivity across various cell populations (Fig. 2A). The proportion of BrdU-labelled cells was increased in CD49b + HR− luminal HU-challenged population compared with unchallenged (Fig. 2D). Only in basal cells was there a modest but statistically significant difference in BrdU-labelled cells between HU-treated genotypes (Fig. 2D and Supplementary Fig. 3b). Overall, the BrdU+ increases occurred in a comparable fashion between wild-type and Brca2$^{mut/WT}$ organoids, especially within the luminal compartment. Thus, transient replicative stress evokes similar cellular phenotypes in the luminal populations, irrespective of genotype.

## Expansion of a Brca2$^{mut/WT}$ HR− luminal population after prolonged replication stress

Replication stress in the breast may be prolonged rather than transient because the breast undergoes many proliferative and remodelling events over its lifetime. We therefore investigated the effects of prolonged exposures to replication stress in Brca2$^{mut/WT}$ mammary organoids. Briefly, organoids (established from up to eight mice per group, aged 9–11 months) were exposed to 1 mM HU for 24 h and allowed to recover for 4 days, before passaging (Fig. 3A). In parallel, we performed equivalent experiments using a physiologically relevant stimulus, 17β-estradiol (E2), which produces oxidative metabolites causing DNA adducts[31]. Organoids were exposed to HU or E2 in this way for five serial passages over 60 days (Fig. 3A). Wild-type and mutant organoids not exposed to HU or E2 were used as controls. Staining for γH2AX after HU exposure confirmed that DNA damage was present at similar levels in all epithelial subpopulations over time (Supplementary Fig. 3a). In both genotypes, basal and HR+ luminal cells showed higher percentages of γH2AX-positive cells compared with HR− luminal cells (Supplementary Fig. 3a), suggestive of lineage-related differences in DNA damage responses and other cellular functioning, as observed in other studies[10,32]. We next assessed whether any differences in cell cycle regulation were observed long term. Apart from an increase in

BrdU positivity in the CD49b+ HR− luminal population at passage 0, the proportion of BrdU-positive cells was not significantly different between the control and HU-treated groups. At passage 0, there was a significant increase in the number of BrdU+ cells compared to passages 2 and 4, in the controls as well as HU- and E2-treated organoids (Supplementary Fig. 3b, c). As organoids were seeded from single cells during passage 0, or by splitting established cultures in passages 2 and 4, we infer that passage 0 organoids may contain a higher frequency of dividing cells than in later passages.

Unchallenged wild-type and Brca2$^{mut/WT}$ organoids retained their regenerative capacity and exhibited consistent organoid growth (Fig. 3B). In contrast, the number of wild-type HU-treated organoids diminished significantly (-four fold reduction, Fig. 3B), indicative of reduced organoid regeneration. Unexpectedly, however, Brca2$^{mut/WT}$ organoids exposed to repeated HU treatments retained regenerative capacity significantly better than HU-treated wild-type organoids. HU-exposed Brca2$^{mut/WT}$ mammary organoids exhibited a statistically insignificant reduction in regenerative capacity compared with unchallenged Brca2$^{mut/WT}$ controls by passage 4. Their regenerative capacity was >three fold higher than that of similarly treated HU-exposed wild-type organoids ($t$ test, $P = 0.056$) (Fig. 3B). There was no measurable difference in regenerative capacity between wild-type or Brca2$^{mut/WT}$ E2-exposed organoids, compared with unchallenged controls (Supplementary Fig 4a), suggesting that estradiol challenge alone does not induce as high a level of replication stress as HU. Collectively, our results suggest that Brca2$^{mut/WT}$ mammary organoids exhibit higher tolerance to replication stress than wild-type organoids.

Copy number alterations (CNAs) have recently been shown to accumulate in non-malignant cells carrying monoallelic *BRCA2* mutations[15]. To evaluate CNAs, we performed shallow whole-genome sequencing (3× coverage) from unchallenged organoids at passage 0 and passage 4 and from HU-exposed organoids at passage 4, using three biological replicates. HU-treated Brca2$^{mut/WT}$ organoids at passage 4 exhibited no increase in detectable CNAs, compared to wild-type organoids or untreated controls (Fig. 3C and Supplementary Fig. 4b, c). We note, however, that random CNAs in organoid systems are likely to be very rare unless enriched non-randomly by selection over prolonged periods, and may therefore elude detection in our experiments. These findings suggest that HU-induced replication stress leads to alterations in the expansion of Brca2$^{mut/WT}$ organoids without inducing severe chromosome instability marked by the accumulation of CNAs, over the timeframe of our experiments.

We next examined the cellular composition of HU-treated organoids. Whole-mount staining for basal (K14) and HR+ luminal (Progesterone Receptor, PR) markers were carried out on three biological samples. Cells not expressing K14 or PR were categorised as HR− luminal. At passage 0, multilineage organoids (containing all three epithelial cell types) were present in 28%, 42%, 24% and 37% of the control wild type, HU-treated wild type (Wild-type HU), control

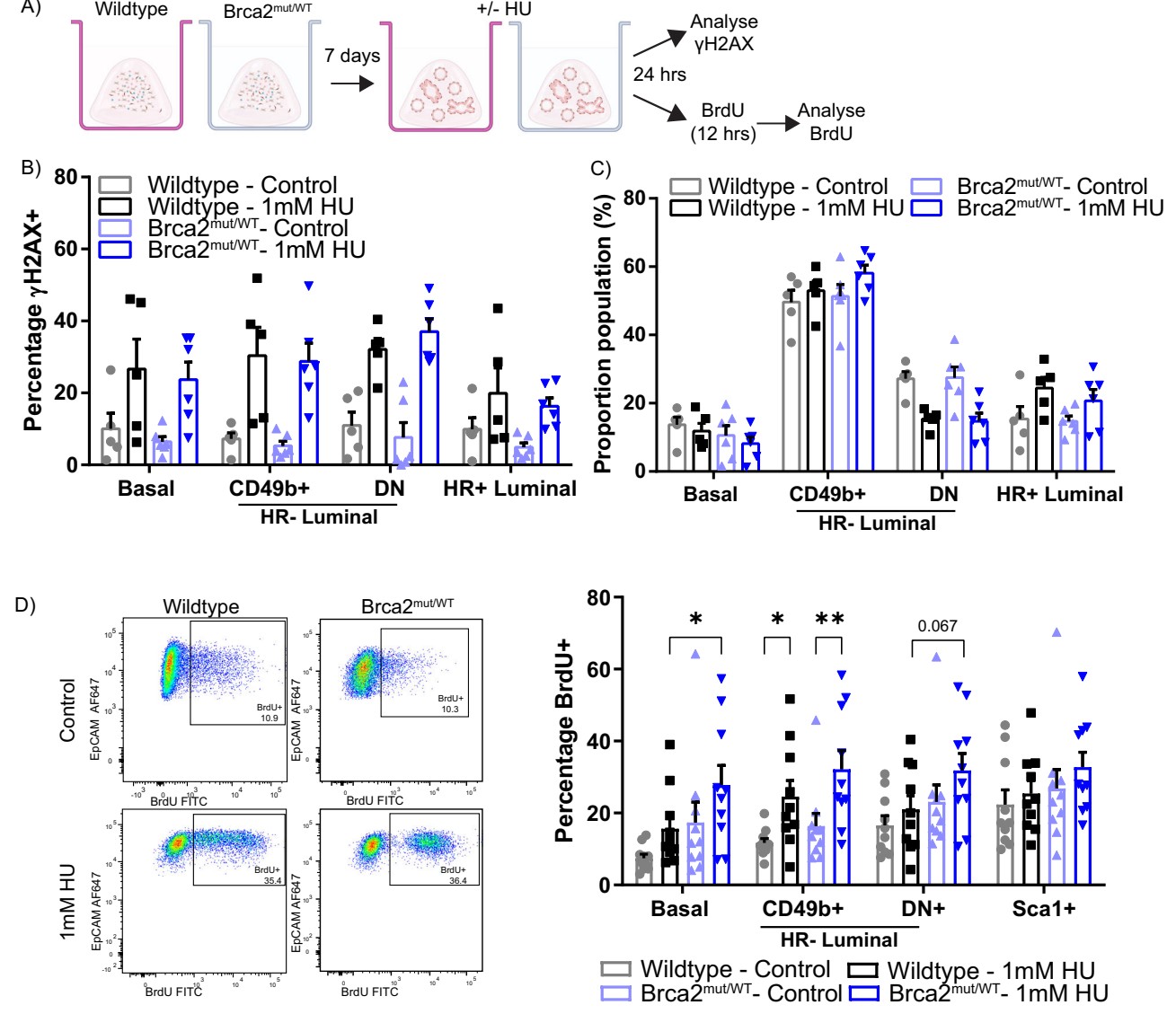

**Fig. 2 | Brca2$^{mut/WT}$ mammary organoids have similar response to short-term DNA genotoxic stress as wild-type mammary organoids. A** Schematic illustrating the experimental pipeline for wild-type or Brca2$^{mut/WT}$ cells grown as organoids, treated with 1 mM HU and assessed for γH2AX or BrdU. Image 'Created with BioRender.com'. **B** Quantification of the percentage of γH2AX+ cells from wild-type or Brca2$^{mut/WT}$ organoids post HU treatment. Data depict the different epithelial populations. Data presented as the mean +/− SEM of $n = 5$ wild-type and $n = 6$ Brca2$^{mut/WT}$ independent biological replicates. Source data are provided. **C** Quantification of the proportion of epithelial populations in the wild-type or Brca2$^{mut/WT}$ organoids post HU treatment. Data presented as mean +/− SEM of $n = 5$

wild-type and $n = 6$ Brca2$^{mut/WT}$ independent biological replicates. **B**, **C** A two-way ANOVA followed by Fisher's LSD test showed no significant differences. **D** Representative flow cytometry analysis of BrdU+ CD49b+ luminal cells (left) and quantification (right) of the percentage of BrdU+ cells in the different epithelial subpopulations from wild-type or Brca2$^{mut/WT}$ organoids post HU treatment. Data presented as mean +/− SEM of $n = 10$ independent biological replicates per group. A two-way ANOVA followed by Fisher's LSD test was performed. Basal: *$P = 0.041$; CD49b+ luminal: *$P = 0.031$, **$P = 0.0078$. Source data are provided as a Source Data file.

Brca2$^{mut/WT}$, and HU-treated Brca2$^{mut/WT}$ groups, respectively (Supplementary Fig. 5a, b). These trends continued in passage 2; however, HU-treated Brca2$^{mut/WT}$ organoids showed a slight increase in HR− luminal cell proportion (Supplementary Fig. 5a, c). Multilineage organoids (containing two or more cell types) were detected in 58%, 69% and 66% of wild-type, Wild-type HU and Brca2$^{mut/WT}$ passage 4 organoids, respectively (Fig. 3D), demonstrating that HU treatments in wild-type mammary cells targeted all cell types equally. Notably, the HU-treated Brca2$^{mut/WT}$ group had significantly reduced proportions of multilineage organoids to 34%, with the remaining 63% containing only the HR- luminal cell type, compared with 29–33% in the other groups (Fig. 3D). Only in passage 4 was there a change following E2 treatments, where the HR- luminal population increased in both wild-type and

Brca2$^{mut/WT}$ organoids (Supplementary Fig. 5d–f). Collectively, our results indicate that prolonged replication stress induced by HU enabled the preferential survival and outgrowth of HR- luminal cells in Brca2$^{mut/WT}$ organoids.

To examine the epithelial subtype proportions, we performed flow cytometry using our standard mammary antibody panels. In concordance with the whole-mount findings, there were minimal changes detected in the proportions of basal and HR+ luminal cell types in any of the conditions tested (Fig. 3E). In contrast, at passage 4, HU-treated Brca2$^{mut/WT}$ organoids had a 20% increase in the proportion of CD49b+ HR- luminal cells compared to the other groups (Fig. 3e). Following E2 treatment, the proportion of basal cells decreased, while the CD49b+ HR- luminal population significantly increased

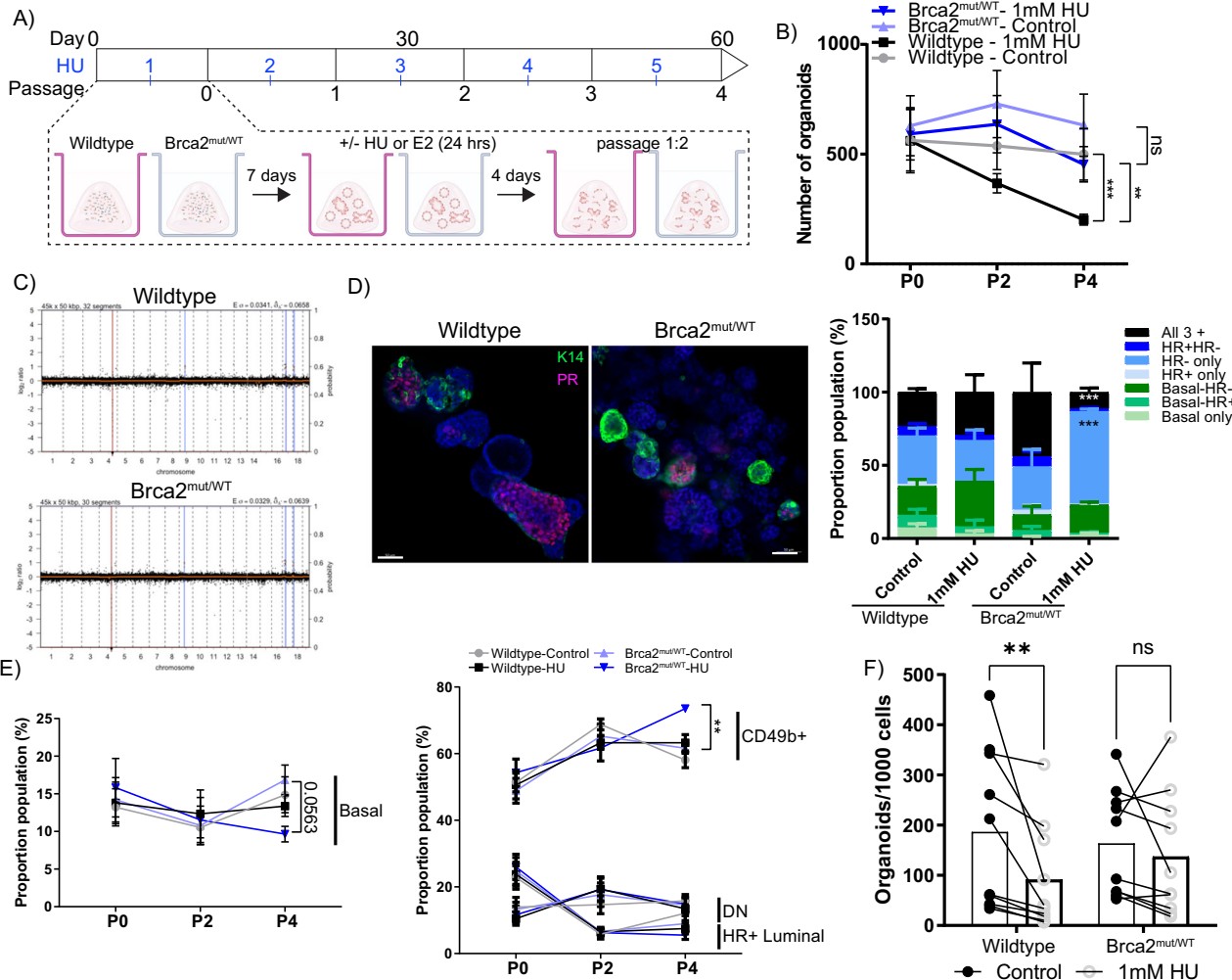

**Fig. 3 | Intermittent genotoxic stress induces expansion of the HR- luminal population in Brca2^mut/WT organoids. A** Schematic illustrating the experimental pipeline for a longitudinal intermittent HU treatment of wild-type or Brca2^mut/WT mammary organoids. Image created with BioRender.com. **B** Quantification of the number of wild-type or Brca2^mut/WT organoids formed over multiple passages (P0, P2 and P4). Data presented as mean +/- SEM (n = 10 per group). Mann–Whitney two-tailed unpaired test was performed, **P = 0.0029. ***P = 0.0005, ns P = 0.056. Source data are provided. **C** aCGH plots from wild-type (upper) or Brca2^mut/WT (lower) passage 4-HU-treated organoids. Somatic CNAs were absent in HU-treated organoids. Representative plots from three independent experiments are shown. **D** Representative immunofluorescent images (left) of wild-type or Brca2^mut/WT organoids post HU treatment after passage 4. K14 (green), PR (magenta) and DAPI (blue). Scale bars, 50 μm. Quantification (right) of each organoid type at passage 4.

Data presented as mean +/- SEM of n = 5 wild-type and n = 5 Brca2^mut/WT independent biological replicates. A two-way ANOVA followed by Tukey's multiple comparison test was performed. Brca2^mut/WT HU all 3 + ***P = <0.0001. Brca2^mut/WT HU HR − only ***P = <0.0001. **E** Quantification of the percentage of basal (left) and luminal (right) cell types formed over multiple passages (P0, P2, P4) from wild-type or Brca2^mut/WT organoids treated with HU. Data presented as mean +/- SEM of n = 8 wild-type and n = 8 Brca2^mut/WT independent biological replicates. Mann–Whitney two-tailed unpaired t test was used. **P = 0.0055. Source data are provided. **F** Quantification of the number of organoids formed from wild-type or Brca2^mut/WT single cells after 5 intermittent HU treatments. Data presented as mean +/- SEM of n = 10 wild-type and n = 10 Brca2^mut/WT independent biological replicates. A two-tailed Wilcoxon matched-pair signed-rank test was performed. **P = 0.00195, ns not significant. Source data are provided as a Source Data file.

(Supplementary Fig. 5g). These changes were detected in both wild type and Brca2^mut/WT, illustrating that the effects of E2 were independent of genotype. We next examined the organoid-forming capacity from HU-treated cells. Passage 4 organoids were dissociated to single cells, reseeded into new organoid cultures, and enumerated after 12 days. All wild-type organoid formation capacity was severely reduced following prolonged HU treatments (Fig. 3F). While Brca2^mut/WT HU-treated cells displayed reduced organoid formation capacity, proliferative ability was not as significantly impaired and three of the eight Brca2^mut/WT samples had a slight expansion following HU treatments. (Fig. 3F). No differences were observed in the E2-treated groups (Supplementary Fig. 5h). Taken together, we show that the Brca2^mut/WT HR- luminal population survives and expands following prolonged replication stress better than other luminal populations and better than wild-type cells.

## Single-cell profiling reveals HR- luminal cell expansion in Brca2^mut/WT mammary cells following replication stress exposure

We next sought to investigate the transcriptional changes associated with the enhanced Brca2^mut/WT organoid survival advantage following repeated replication stress. We performed single-cell (sc)RNA-seq from early (passage 0) and late (passage 4) organoid culture time points. Across the four different conditions (wild-type or Brca2^mut/WT with or without HU treatments) and time points, a total of 34,200 cells were retained for analysis following quality controls (see "Methods"). Unsupervised clustering across all high-quality cells and visualising the results using uniform manifold approximation and projections (UMAP) revealed five transcriptionally distinct subclusters (Fig. 4A). Lineage-specific markers were examined to identify the different clusters, and cell types were labelled using these marker genes[24,33]. The dominant clusters were the HR- luminal population, which consisted of

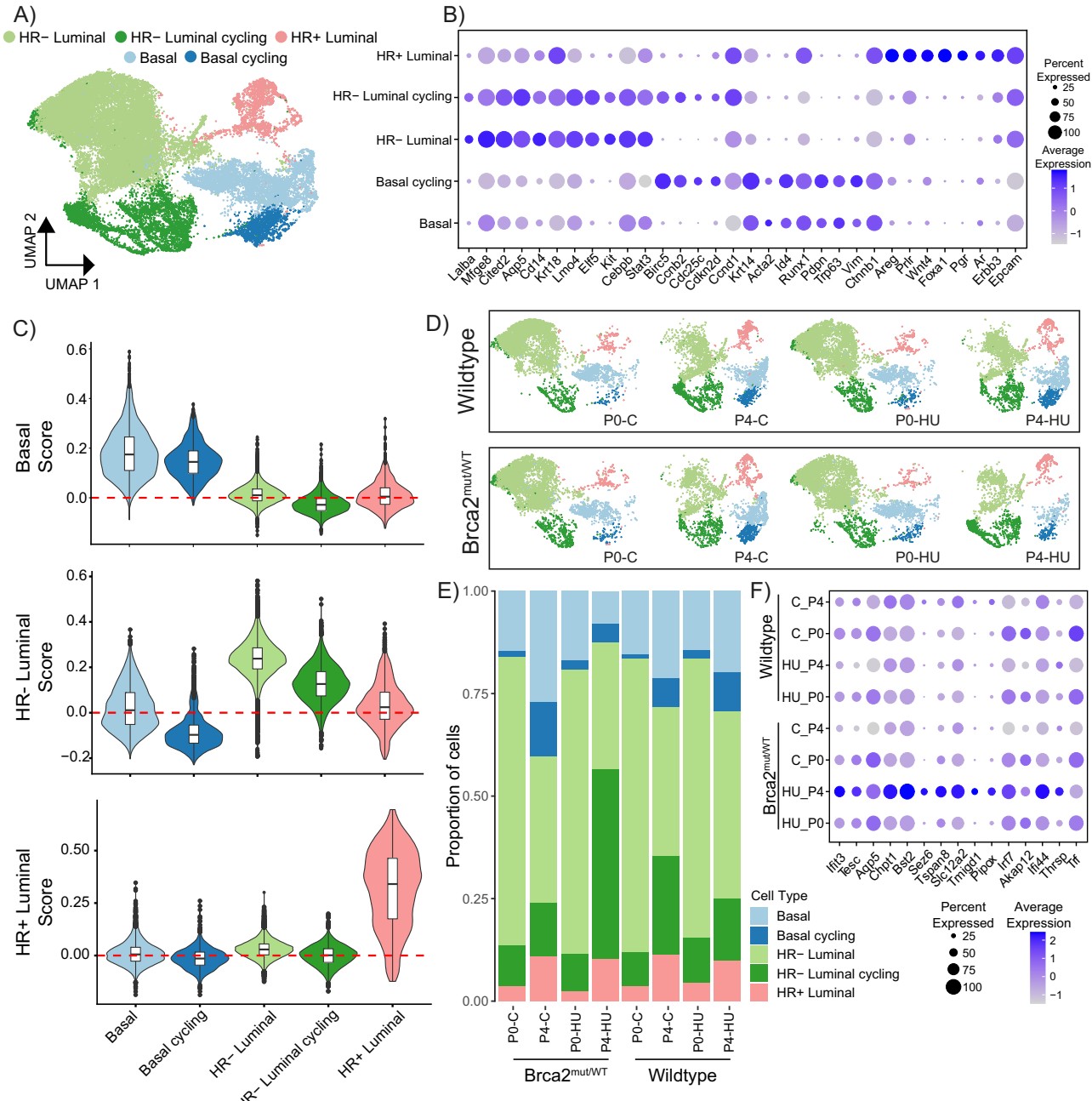

**Fig. 4 | Single-cell transcriptomics reveal an expanded HR− luminal cycling population in Brca2[mut/WT] organoids. A** Projection of dimensionality reduced (UMAP) scRNA-seq data (left, *n* = 34,200 cells in dataset from wild-type or Brca2[mut/WT] organoids generated from *n* = 3 individual mice for each condition (Passage 0-Control, Passage 0-HU, Passage 4-Control, Passage 4-HU)) coloured by cell type. **B** Dot plot showing the expression of established mammary and proliferating cell markers associated with each cluster. Marker genes (*x* axis) for each major cell type cluster (*y* axis). Circle size indicates the percent of cells in the cluster expressing the gene. Filled colour represents the normalised and scaled mean expression level. **C** Violin plots of basal (upper), HR− luminal (middle) and HR+ luminal (bottom) cell scores across the mammary organoid cell clusters (data are presented as median and quartile from wild-type or Brca2[mut/WT] organoids generated from *n* = 3 individual mice). **D** UMAP analysis control (C) or HU treatment (HU) organoids at passage 0 (P0) or 4 (P4) wild-type or Brca2[mut/WT] organoids; cell types are coloured. **E** Frequency of cell types for each group, colour coded by cell type. **F** Dot plot showing the expression of 15 of the top 30 differentially expressed marker genes in Brca2[mut/WT] P4 HU-treated organoids compared to the other groups. Circle size indicates the percentage of cells in which the gene expression was detected. Fill colour depicts the normalised and scaled mean expression level.

cells expressing known HR− luminal genes (*Lalba, Mfge8, Cd14, Elf5, Kit*), and a HR− luminal cycling population, which expressed not only HR- luminal genes but also known cell cycle genes including *Birc5, Ccnb2, Cdc25c* and *Cdkn2d*. A basal population was identified that expressed known basal markers (*Krt14, Id4, Pdpn, Tp63, Vim*), and a basal cycling population was also identified. The final cluster was identified to contain HR+ luminal cells expressing genes including *Areg, Prlr, Foxa1, Pgr* and *Ar* (Fig. 4B).

After annotating individual lineage-specific genes, we cross-examined published mammary epithelial gene signatures derived from sorted HR- luminal, HR+ luminal, and basal cells[34] to elucidate the extent to which our organoids could maintain lineage-specific transcriptional programmes. We scored each epithelial cluster from the organoids with respect to each lineage gene signature (Fig. 4C). We found that mammary organoids faithfully recapitulated the gene expression signatures of primary mammary tissues. The basal gene

score was highest in the organoid basal and basal cycling clusters, the HR− luminal score was highest in the organoid HR− luminal and HR− luminal cycling clusters, and the HR+ luminal score was highest in the organoid HR+ luminal cluster (Fig. 4C). UMAPs of wild-type and Brca2[mut/WT] organoids resulted in all epithelial clusters being maintained (Fig. 4D). Reassuringly, both unchallenged and HU-treated wild-type and Brca2[mut/WT] organoids at passage 0 and 4 showed the same gene signature correlations for the basal score (Supplementary Fig. 6a), HR− luminal score (Supplementary Fig. 6b), and the HR+ luminal score (Supplementary Fig. 6c). These findings demonstrate that mammary organoid cultures maintain the different lineages throughout the culture period irrespective of genotype or replication stress. Consistent with whole-mount immunofluorescent analysis (Fig. 3D), HU-treated Brca2[mut/WT] organoids at passage 4 had the largest proportion of cells within the HR- luminal clusters, with the dominant population being the HR- luminal cycling cluster compared to unchallenged Brca2[mut/WT], or wild-type organoids (Fig. 4E).

We then examined differences in gene transcription in HU-treated Brca2[mut/WT] cells compared to unchallenged Brca2[mut/WT] and wild-type cells (+/− HU treatments) at passage 4. Among the top 30 differentially expressed genes, interferon-related genes (*Ifit3, Irf7, Ifi44*) and luminal alveolar lineage-specific genes (*Tspan8, Thrsp, Aqp5*)[20,24,33] were elevated in the HU-treated Brca2[mut/WT] (Fig. 4F and Supplementary Data 1). These genes were highly expressed in the HU-treated Brca2[mut/WT] samples from passage 4, and their expression increased over time (Fig. 4F). Taken together, we showed that repeated stress events affect Brca2[mut/WT] mammary epithelial cells differently. Basal cells reduced in number while HR− luminal cells significantly expanded. This expanded HR- luminal population also showed upregulation of interferon-related and alveolar transcripts.

## HR− luminal cells activate an interferon response after replication stress

To examine the mechanistic impact of replication stress on the luminal compartment that may confer growth advantage, we performed unsupervised clustering across the HR- luminal clusters, resulting in nine transcriptionally distinct subclusters (Fig. 5A, B). At passage 4, the HU-treated Brca2[mut/WT] organoids had the highest proportion of cells in cluster 4 compared with unchallenged Brca2[mut/WT], or wild-type organoids (Supplementary Fig. 7a). As cluster 4 was annotated as the HR-luminal cycling cluster, we next investigated the biological terms associated with this cluster. The GO specific to cluster 4 genes in the HU-treated Brca2[mut/WT] organoids were significantly attributed to Type 1 interferon responses (Fig. 5C). Similarly, clusters 1, 2 and 3 in the HU-treated Brca2[mut/WT] organoids also showed enrichment in Type 1 interferon and Neutrophil-related response genes (Supplementary Fig. 7b–d and Supplementary Data 2). Genes associated with Interferon response terms such as *Ifit3, Ir7, Ifit1, Blc2* and *Stat1* showed increased expression in the HU-treated Brca2[mut/WT] luminal cells compared with the other conditions, especially after prolonged exposures to replication stress (Fig. 5D). Elevated terms in the Neutrophil-related response in clusters 1 and 3 included genes such as *Slpi, Fabp5, Nme2, Ptx3* and *Tmsb4x* (Supplementary Data 2). MSigDB and KEGG gene enrichment analysis on clusters 2 and 4 corroborated the GO terms for these clusters, while cluster 3 analysis showed upregulation of epithelial-to-mesenchymal transition-related genes (Supplementary Fig. 7e–g and Supplementary Data 2). We further investigated the cGAS-Sting pathway, using genes detailed in the Cytosolic DNA-sensing pathway from mmu04623 KEGG database, and observed an overall elevation in the expression of genes within this pathway in passage 4 organoids compared to passage 0 (Supplementary Fig. 7f). As cGAS-Sting pathway genes were activated in all passage 4 conditions, suggests that the cGas-Sting pathway was not exclusively activated in HU-treated Brca2[mut/WT] organoids (Supplementary Fig. 7f). Cluster 5 was increased in the HU-treated wild-type group, and when compared with the wild-type controls, it displayed an

elevation in DNA damage repair pathways (Supplementary Fig. 7h and Supplementary Data 3). The wild-type control group had increased proportions of clusters 1 and 4 with elevated terms related to neutrophil response and endocytosis (Supplementary Fig. 7i–j and Supplementary Data 3), suggesting that the interferon response is mainly associated with Brca2[mut/WT] organoids. Induction of immune-associated responses, including the genes we identified, have been reported in biallelic BRCA2-mutant cancer cells[35]. However, these interferon responses have not previously been demonstrated in mammary epithelial cells bearing monoallelic BRCA2 mutations. Our findings suggest that the expression of interferon response genes in Brca2[mut/WT] HR- luminal cells is induced by persistent replication stress.

To test the role of these interferon-related genes in organoid survival following replicative stress, we pre-treated passage 4 organoids for 3 days with recombinant Interferon β (IFNβ; inducing a type 1 interferon response) or a Janus Kinase inhibitor (AZD1480/Jaki; blocking Stat1 activation downstream of interferons) before subjecting the organoids to HU or E2 treatments. IFNβ treatment suppressed proliferation in the HU-treated organoids (Fig. 5E) and affected Brca2[mut/WT] organoid size (Fig. 5F), suggesting IFN signalling may provide an anti-apoptotic or pro-proliferative survival advantage. However, there were no changes in proliferation or organoid size following IFNβ exposure in the E2-treated organoids (Supplementary Fig. 8a, b). Conversely, AZD1480/Jaki did decrease organoid size in all conditions (Fig. 5F and Supplementary Fig. 8b). These results suggest that the interferon response contributes to the survival advantage of Brca2[mut/WT] organoids following replicative stress.

The dominant cluster 4 population in the Brca2[mut/WT] group is annotated as the HR- luminal cycling population, and exhibits high expression of mammary genes such as *Tspan8*[36] and *Thrsp*[24] (Fig. 4F). As Tspan8 is a cell surface marker highly expressed in this cluster, we utilised it as a surrogate marker to identify this cluster and thus determine whether HR- luminal cells in this cluster were being depleted following various treatments. Whilst we could detect Tspan8-positive luminal cells in all IFNβ-treated organoids, no changes in Tspan8 expression were observed in HU- or E2-treated organoids (Fig. 5G and Supplementary Fig. 8c). As IFNβ and Jaki treatments reduced proliferation and organoid growth, respectively, following HU treatments, we next assessed whether cells were able to recover from these treatments and proliferate. To examine proliferative potential after treatments, we reseeded single cells into new organoid cultures. In IFNβ-treated groups, the cells recovered from growth inhibition to form comparable number of organoids to controls (Fig. 5H and Supplementary Fig. 8d). While Jaki-treated groups showed an increase in organoid-forming capacity, this was not significant (Fig. 5H). Interestingly, Jaki treatments in the E2-exposed Brca2[mut/WT] group showed a decrease in Tspan8-positive cells and organoid numbers compared to the Brca2[mut/WT] control (Supplementary Fig. 8c, d), suggesting that E2 exposure minimised the response to Jak inhibition. Flow assessment of the endpoint organoids demonstrated that HU-treated Brca2[mut/WT] organoids had a higher proportion of Tspan8+ cells following IFNβ withdrawal, whereas HU-treated wild-type organoids exhibited a reduction (Fig. 5I), and no changes were observed in the E2-treated groups (Supplementary Fig. 8e). In summary, replication stress induced a type I interferon response selectively in Brca2[mut/WT] HR− luminal cells, prompting a cell survival response via entering a non-proliferative state for DNA repair and/or escape from cell toxicity. This response follows activation of the Jak pathway, and increased Tspan8 expression allows a luminal cycling cell cluster to persist, enabling the HR− luminal cells to maintain proliferation and growth.

## Deletion of HR− luminal cycling cluster genes compromise survival after replication stress

Our findings indicate that prolonged HU exposure caused an expansion of a HR− luminal cycling cluster. Overlay plots showed enrichment

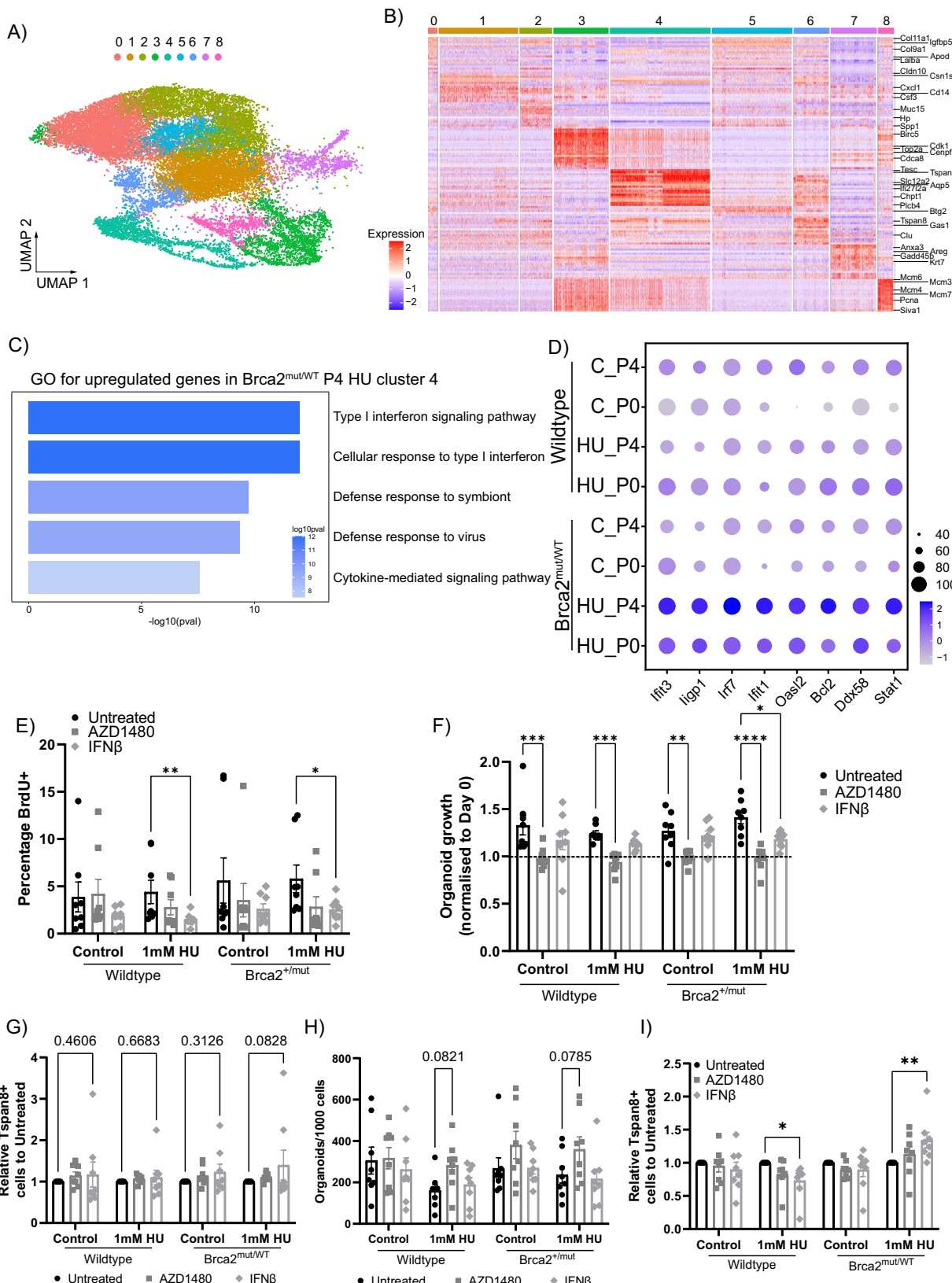

in the luminal cycling cluster 4 population, which contain high expression of *Tspan8* (Fig. 6A, B). To functionally validate whether Tspan8 positivity increases in HU-treated Brca2[mut/WT] cells, fresh mammary cells from seven biological samples were seeded at passage 0, and the longitudinal assay was repeated. We observed Tspan8-expressing cells expanded in the HR− luminal cells of HU-treated

Brca2[mut/WT] organoids, compared with unchallenged Brca2[mut/WT] or wild-type (+/− HU treatment) organoids (Fig. 6C). As E2 exposure expanded the HR- luminal population, we assayed organoids exposed to E2 to determine whether Tspan8 was comparably increased, and we observed Tspan8-positive cells were increased in the HR- luminal population, albeit not selective to Brca2[mut/WT] genotype

Fig. 5 | **Investigating the HR- luminal cycle population shows increased Type I interferon response. A** Distribution of nine HR− luminal epithelial cell subtypes on the UMAP. **B** Heatmap of the normalised top 20 DEGs of HR- luminal clusters from all organoid groups (red = high expression, blue = low expression). **C** The top 5 GO Biological terms associated with the upregulated genes in Brca2[mut/WT] HU passage 4 cluster 4 cells (a Wilcoxon rank-sum test was performed). **D** Dot plot showing the expression of the genes identified in (**C**) in Brca2[mut/WT] P4 HU-treated organoids compared to the other groups. Circle size indicates the percentage of cells in which the gene expression was detected. Fill colour depicts the normalised and scaled mean expression level. **E** Quantification of the percentage of BrdU+ cells in the HR− luminal subpopulations from wild-type or Brca2[mut/WT] passage 4 organoids after 4 days of IFNβ or AZD1480 treatment. Data presented as mean +/− SEM of $n = 8$ wild-type and $n = 8$ Brca2[mut/WT] independent biological replicates. A two-way ANOVA followed by a Fisher's LSD test was performed, *$P = 00.038$, **$P = 0.0067$. **F** Quantification of average organoid size after 4 days of INFβ or AZD1480 treatment normalised to day 0 treatment organoid size. Dashed line indicates day 0 area. Data presented as mean +/− SEM of $n = 8$ wild-type and $n = 8$ Brca2[mut/WT]

independent biological replicates. A two-way ANOVA followed by a Dunnett's multiple comparison test, *$P = 0.0109$, **$P = 0.0015$ and ***$P = 0.0007$, ****$P < 0.0001$. **G** Quantification of the percentage of Tspan8+ cells in the HR− luminal population after 4 days of IFNβ or AZD1480 treatment on passage 4 wild-type or Brca2[mut/WT] organoids. Data presented as the mean +/− SEM of $n = 8$ wild-type and $n = 8$ Brca2[mut/WT] independent biological replicates. A two-way ANOVA followed by a Fishers LSD test showed no statistical differences. **H** Quantification of the number of organoids formed from wild-type or Brca2[mut/WT] passage 4 single cells after IFNβ or AZD1480 treatments. Data presented as mean +/− SEM of $n = 8$ wild-type and $n = 8$ Brca2[mut/WT] independent biological replicates. A two-way ANOVA followed by a Fishers LSD test showed no statistical differences. **I)** Quantification of the percentage of Tspan8+ cells in the HR- luminal populations of wild-type or Brca2[mut/WT] organoids from **H**. Data presented as the mean +/− SEM of $n = 8$ wild-type and $n = 8$ Brca2[mut/WT] independent biological replicates. A two-way ANOVA followed by a Fishers LSD test, *$P = 0.0172$, **$P = 0.0029$. Source data are provided as a Source Data file.

(Supplementary Fig. 9a). Next, we sought to identify whether Tspan8 was essential for Brca2[mut/WT] organoids to sustain a growth advantage following replicative stresses. To genetically modify the organoids, we adopted the RNP CRISPR approach, where Cas9 protein and synthetic single-stranded gRNA complexes were transiently introduced via nucleofection into the cells. This strategy has the advantage that the Cas9-gRNA complexes are rapidly degraded, thus minimising off-target effects compared to plasmid-based approaches[37,38].

Given that *Tspan8* was highly expressed in cluster 4 (Fig. 6B) and a large proportion of the HR− luminal populations (Supplementary Fig. 9b), we tested the role Tspan8 may play in HR− luminal survival and expansion under replication stress. We generated Tspan8-knockout (Tspan8 ko) mammary cells from primary 2D cultures and seeded cells into organoid cultures. Tspan8 ko editing efficiency was confirmed by DNA sequencing, and protein depletion was validated via flow cytometry analysis (Fig. 6D). Tspan8 expression was detected in unedited wild-type and Brca2[mut/WT] organoids at passage 3, as expected, while Tspan8 ko organoids maintained loss of Tspan8 expression both at the beginning and after prolonged (1.5 months) organoid culture (Supplementary Fig. 9c). Both wild-type and Brca2[mut/WT] Tspan8 ko organoids exhibited a reduction in, but not complete absence of, the HR- luminal progenitor population (CD49b+ cells, Fig. 6E). To determine whether Brca2[mut/WT] organoids deficient in Tspan8 were able to survive replication stress, we repeated the longitudinal HU intermittent assay. The number of organoids formed during passage 3 from unchallenged Tspan8 ko wild-type or Brca2[mut/WT] organoids were equivalent to non-edited (Fig. 6F). As expected, prolonged replication stress reduced the number of wild-type organoids regardless of Tspan8 ko, indicating that wild-type organoids are less affected by Tspan8 loss (Fig. 6F). Tspan8 ko nearly eliminated the regenerative capacity of HU-treated Brca2[mut/WT] organoids (Fig. 6F), reverting the replicative stress response akin to wild-type organoids.

To assess whether other genes expressed in cluster 4 contribute to this response, we examined our scRNA-seq data and found that *Thrsp*, a gene expressed in luminal alveolar cells[20,24], was also highly expressed in the HR- luminal cycling cluster 4 population (Supplementary Fig. 9d). We generated Thrsp knockout (Thrsp ko) organoids following the RNP CRISPR pipeline and observed a consistent disruption of *Thrsp* via a 1 base pair out-of-frame insertion (Supplementary Fig. 9e). Thrsp ko organoids phenocopied Tspan8 ko organoids in exhibiting a reduction in HR- luminal population in the Brca2[mut/WT] organoids, with minimal impact on wild-type populations (Supplementary Fig. 9f). The longitudinal HU intermittent exposure assay confirmed that Thrsp ko in wild-type organoids did not alter organoid growth dynamics, as similar number of organoids formed compared to non-edited organoids (Supplementary Fig. 9g). Similar to the response in Tspan8 ko organoids, Thrsp ko Brca2[mut/WT] organoids were sensitive

to replication stress and significantly reduced organoid numbers (Supplementary Fig. 9g). Taken collectively, our findings suggest that genetic deletion of *Tspan8* or *Thrsp* genes, whose expressions are elevated in cluster 4, are required for maintenance of the HR- luminal cycling population and that disruption of the HR- luminal cycling cluster 4 population significantly impairs the response to prolonged replication stress.

To extend these findings, we investigated *TSPAN8* and *THRSP* expression among the germline BRCA2-mutant tumours in the TCGA breast cancer dataset. The data showed an increase in *TSPAN8* expression, but not *THRSP* expression, in BRCA2-mutant tumours compared with wild-type tumours (Supplementary Fig. 10a). However, both *TSPAN8* and *THRSP* expression were lower in BRCA2-mutant and non-BRCA2-related cancers when compared to normal tissue (Supplementary Fig. 10a, b). We surmise that *Tspan8/Thrsp* upregulation could be an early response which is not sustained during tumour progression. However, because elevated *Tspan8/Thrsp* expression was accompanied by an interferon response in our model, we next investigated whether BRCA2-mutant cancers displayed increased interferon response pathway expression. We analysed tumours that contain either germline or somatic BRCA2 mutations, on one or both alleles, to provide a sufficient sample size. We find that BRCA2-mutant tumours upregulate interferon pathway genes (Supplementary Fig. 10c), in line with our findings and previous reports[39], suggesting this response is conserved between mouse models and human tumour data.

## Discussion

Cells heterozygous for pathogenic BRCA2 mutations do not apparently exhibit baseline phenotypes associated with loss of function in DNA repair. How heterozygous BRCA2 mutations affect epithelial cell lineages, which may predispose these cells towards the development of cancer, remains poorly understood. Here, we demonstrate that Brca2[mut/WT] mammary organoids are not innately primed for aberrant behaviour, yet prolonged exposure to replication stress results in a preferential expansion of HR− luminal cells. This expanded population expressed genes characteristic of proliferation, type I interferon response, and elevated mammary HR− alveolar genes (such as *Thrsp* and *Tspan8*) for sustained survival. Inactivation of these HR- alveolar genes reversed Brca2[mut/WT] organoid survival response to replication stress.

Our data showing Brca2[mut/WT] HR- luminal cell expansion raise important questions about the aetiology of BRCA2-associated breast cancers. While BRCA1 mutation carriers have an elevated risk for triple-negative breast cancers (TNBCs), BRCA2 mutation carriers develop breast cancers of different subtypes at a similar frequency to the general population and largely develop HR+ tumours[2,40]; nevertheless, ~20% of BRCA2-associated tumours are TNBCs[41]. Luminal HR+ tumours

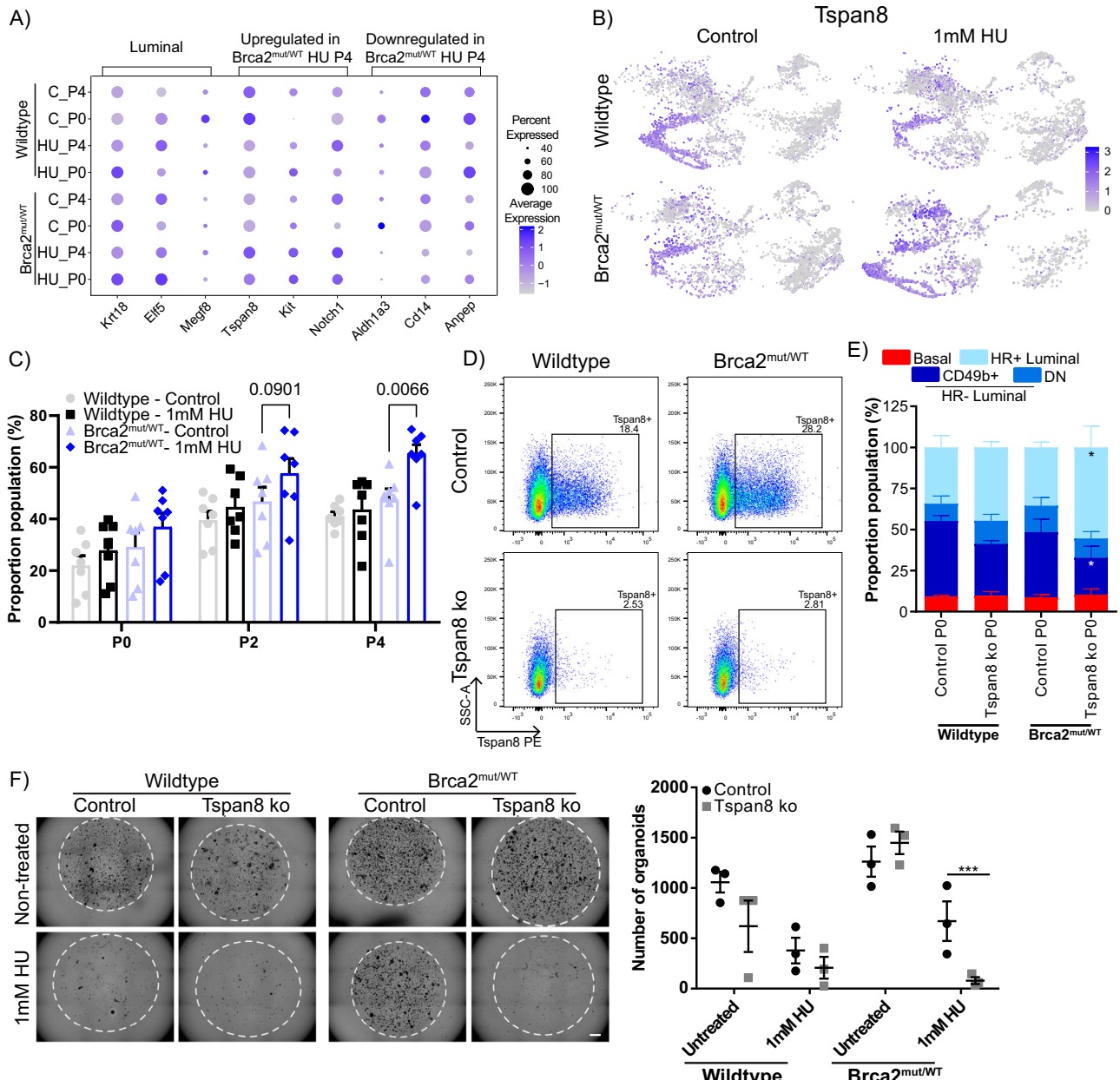

**Fig. 6 | Deletion of Tspan8 expression in the HR- luminal cells render Brca2^mut/WT organoids incapable of recovery from DNA damage. A** Dot plot showing the expression selected HR- luminal expressed marker genes upregulated or downregulated in Brca2^mut/WT P4 HU-treated organoids compared to the other groups. Circle size indicates the percentage of cells in which the gene expression was detected. Fill colour depicts the normalised and scaled mean expression level. **B** UMAP with overlaid expression of Tspan8 gene (purple) from control or HU-treated passage 4 wild-type or Brca2^mut/WT organoids. **C** Quantification of the percentage of Tspan8+ cells in the luminal progenitor population over different passages (P0, P2 and P4) from wild-type or Brca2^mut/WT organoids treated with HU. Data presented as the mean +/− SD (*n* = 7). A two-way ANOVA followed by a Fishers LSD test was performed. **D** Representative flow cytometry analysis of Tspan8

expression from control (upper) or Tspan8-knockout (ko, lower) wild-type and Brca2^mut/WT organoids. **E** Quantification of the percentage of epithelial cell types from control or Tspan8 KO wild-type or Brca2^mut/WT organoids. Data presented mean +/− SD (*n* = 3). A two-way ANOVA followed by a Fishers LSD test, Brca2^mut/WT Tspan8 KO CD49b+ **P* = 0.0441, Brca2^mut/WT Tspan8 KO Sca1+ **P* = 0.0206. **F** Representative brightfield images of Brca2^mut/WT organoids derived from control or Tspan8 KO cells at passage 3 after 4 intermittent HU treatments (left). Scale bar = 800 μm. Quantification (right) of the number of wild-type or Brca2^mut/WT organoids from control and Tspan8 KO post HU treatments at passage 3. Data presented mean +/− SEM (*n* = 3). A two-way ANOVA followed by a Tukey's multiple comparison test, ***P* < 0.001. Source data are provided as a Source Data file.

more closely resemble HR+ mammary epithelial cells[12,20,27]. In contrast, TNBCs are thought to arise from HR- luminal cells[14,42–44], and premalignant changes identified in the breasts of BRCA1 mutation carriers mostly occur in this cell type[12,45,46]. That we observe an expansion of HR- luminal cells after HU treatment in our organoids suggests our models may favour cellular advantage towards a HR− phenotype, although it remains to be seen whether these expanded populations

contribute towards tumorigenesis. Thus, the mechanisms of transformation giving rise to HR− and HR+ tumours in BRCA2 mutation carriers may be distinct.

Non-cancerous BRCA1-mutant breast tissues and mammary glands have been reported to exhibit an expansion of the HR- luminal progenitor cell population[12,13,44]. Aberrant luminal programmes were evident in pre-malignant *Brca1/Trp53* LOF HR- luminal progenitors[44],

where induction of environmental DNA damage may activate hormone-independent luminal progenitor programmes prior to tumour development[43]. These studies suggest that intact DNA repair mechanisms may maintain a balance of cell differentiation programmes required for normal luminal cellular function. However, in the absence of functional DNA repair mechanisms (i.e., mono- or biallelic loss of Brca1) this leads to an imbalance of differentiation programmes, increased progenitor populations, and resultant aberrant behaviours.

We have demonstrated insights into the epithelial cellular response to replication stress in Brca2-mutant mammary tissues. These insights suggest that mammary luminal cells undertake distinct aberrant behaviours prior to malignant transformation. These distinct pathways may cause age-related expansions in different cell subtypes. Age of onset in TNBCs was higher in younger patients when compared with ER+ tumours in a cohort of pathogenic BRCA2 mutation carriers[47]. HR− cell expansion has recently been reported in non-malignant epithelium derived from the breast tissues of BRCA2 mutation carriers[13], consistent with our work. Within the timeframe of our models, we observed a phenotype in the Brca2$^{mut/WT}$ HR- luminal cell response following replication stress. Although we followed the organoids for a total of 60 days, we cannot exclude that Brca2$^{mut/WT}$ HR+ luminal cells may exhibit an aberrant response to replication stress over a longer timeframe.

Activation of an intrinsic interferon response has been observed in BRCA2-depleted breast cancer cells and has been attributed to endogenously arising DNA damage because these cells are defective in DNA repair[35,48]. However, our data indicate that the interferon response to replication stress may be a cell-intrinsic phenotype in HR− luminal cells, which prompts them to enter a non-proliferative state that may promote DNA repair and/or escape from cell toxicity. This intrinsic interferon response includes a set of genes within the IFN-related DNA damage resistance signature (IRDS) (including STAT1, IRF7, IFIT1/3, IFITM1, ISG15 and OAS family, and BST2 –found in this study)[49]. Breast cancer cells with constitutive IRDS fail to transmit cytotoxic signals, resulting in pro-survival signals[49] and correlating with resistance to radiotherapy and chemotherapy in breast cancers[50], but relatively little is known about how this pathway functions to facilitate the progression to tumorigenesis. Our data show increased expression of IRDS at the passage 4 time point, implicating activated IRDS in driving a pro-survival signal in Brca2$^{mut/WT}$ HR- luminal cells. Upregulation of interferon and related genes was also observed in human BRCA2-mutant tumour samples. Thus, an interferon response may be a survival advantage in Brca2-mutant mammary cells. Emerging evidence suggests Brca2-mutant tumours activating immune and interferon-related signalling programmes[35,48] and respond to checkpoint blockade immunotherapies[51]. Future studies could seek to elucidate the role of cell-intrinsic interferon signalling in aberrant cell formation and to determine the precise damage signal which activates this response.

In addition to the interferon response, we show that Brca2$^{mut/WT}$ HR- luminal cells upregulate various of their lineage marker genes after repeated HU exposure and that alveolar mammary genes such as *Tspan8* and *Thrsp* functionally enable a growth advantage in Brca2$^{mut/WT}$ cells. In the mammary gland, high Tspan8 expression in myoepithelial cells identified a quiescent mammary stem cell phenotype, which became activated by ovarian hormones[36]. In this study, we observe increased Tspan8-positive luminal cells, and upon depletion of Tspan8, HU-treated Brca2$^{mut/WT}$ cells succumbed to replication stress, demonstrating both a dependence on *Tspan8* to stimulate the proliferative reservoirs necessary for cell survival and a bona fide functional role of *Tspan8* in HR− luminal cells. However, these results do not rule out the possibility that the observed effect of Tspan8 depletion could also arise from it's effects on basal or HR+ luminal cells. Increased cell survival was also dependent on *Thrsp* expression.

Thrsp stimulates fatty acid synthase (FASN)[52], and increased levels of FASN have been shown to increase resistance to replicative drugs in vitro[53]. In our study, we also identified a interaction between interferon responses and the mammary HR- luminal cells lineage programme (including *Tspan8* and *Thrsp*). Future work to interrogate the expansion of HR- luminal cells in a prolonged replicative stress environment will determine the combined involvement of an interferon response and amplified lineage programme in the emergence of transformative phenotypes.

In summary, we have established a murine organoid model to investigate the behaviour of mammary epithelial cells carrying germline monoallelic Brca2 mutations under replication stress. DNA replication stress in the breast could arise naturally from infections, the differentiation process during pregnancy/lactation, or remodelling throughout the mammary gland's life cycle. In our model, unchallenged murine monoallelic Brca2-mutant mammary cells are not innately primed for survival behaviour. Only following multiple challenges do the HR− luminal Brca2$^{mut/WT}$ cells exhibit a survival advantage, mediated by the combined and possibly stepwise effects of IFN responses and alveolar mammary gene expression, selectively expanding a luminal cycling population. Taken together, our data suggest, based on our mouse organoid model, that this expanded luminal population may represent Brca2-mutant cells that have activated a survival programme and are adapted to endure replication stress; in turn, this population may contribute to mammary carcinogenesis.

## Methods

### Mice
Mammary glands from Brca2 germline and wild-type mouse strains were provided by the Venkitaraman group. Tissues collected from virgin adult young (3-month-old) and aged (7–11-month-old) female Brca2 Tr/WT[22] and littermate Brca2 WT/WT mice. All mice were treated in strict accordance with the local ethical committee (University of Cambridge Licence Review Committee) and the UK Home Office guidelines. Animals were housed in a pathogen-free environment with ab libitum to diet and sanitised water. Mice were housed in a standard facility and maintained under standard conditions: 20–24 °C, 40–70% humidity and a 12 h light/dark cycle.

### Organoid cultures

**Mammary organoids.** Mammary epithelial cells were collected from third and fourth mammary glands of virgin mice. Briefly, lymph nodes were removed, and glands minced with surgical scissors before enzymatical dissociation for 1.5 h in DMEM/F12 (1:1) Supplemented with 2 mg mL$^{-1}$ collagenase (Roche) + Gentamicin (Gibco). Samples were briefly vortexed every 30 min. Mammary gland fragments were treated with NH$_4$CL to lyse red blood cells, then dissected to single cells with 0.05% Trypsin-EDTA (STEMCELL Technologies) and 5 mg ml$^{-1}$ dispase (STEMCELL Technologies) and 1 mg ml$^{-1}$ DNAse (Sigma) and filtered through a 40-μm cell strainer (Falcon).

Single cells were mixed with Matrigel (Corning) and seeded (5500 cells per 35 μl Matrigel) in 24-well plates. In all, 500 μl of MammoCult™ Organoid Growth Medium (Mouse, STEMCELL Technologies) + Gentamicin (Gibco) was added per well and medium was replenished every other day. Cultures were maintained in a 37 °C humidified atmosphere under 5% O$_2$, 5% CO$_2$ for 10–12 days. For replicative or oxidative stress experiments: Hydroxyurea (1 mM, unless otherwise indicated, Sigma) or Oestrogen (1 μM, Sigma) was added to organoid cultures on day 5/6 for 24 h. Cultures washed with PBS and replenished with fresh medium. For inhibitor experiments: Recombinant Interferon β (250U/ml, Peprotech) or Jak inhibitor (AZD1480 100 nM; Calcembio) was added to passage 4 organoids on day 5 for a total of 4 days. Replicative stress or oxidative stress was performed on day 7 by administering 1 mM HU or 1 μM Oestrogen, respectively, to the organoids for 24 h.

Single cells were used for the initial culture to enable equal cell numbers seeded into the Matrigel domes, which can be accurately quantified. Multiple domes (6–11) were set up for each genotype and treatment groups (control, HU or Oestrogen) to enable sufficient material for downstream assays and passaging.

**Organoid passaging.** Mammary organoids were maintained in culture by passaging them every 10–12 days at a 1:2 ratio split. Briefly, mammary organoids were washed by PBS and mechanically released from the Matrigel by breaking the matrix with a P1000 pipette and treated with TrypLE (Invitrogen) for 2–3 min at 37 °C to generate fragments. Following washes with trypsin inhibitor, mammary fragments were centrifuged at $100 \times g$ for 5 min at 4 °C. Supernatant containing single cells was removed and the fragments were resuspended in Matrigel, seeded in 24-well plates, and exposed to previously described culture conditions. For quantitative organoid formation analysis: organoids were mechanical dissociated in TrypLE for 10–15 min at 37 °C to generate a single-cell suspension. Single cells were resuspended in Matrigel and cultured. Organoids were imaged on a SX5 Incucyte® (Sartorius). Organoids were imaged at the start and end of every passage. For inhibitor experiments: organoids were imaged at the start and end of treatments. Organoid numbers were calculated using the Incucyte® organoid analysis software module (Sartorius).

CRISPR-edited cells: 4000 single cells were mixed with 35 μl Matrigel in 24-well plates and Supplemented with 500 μl of Mammo-Cult™ Organoid Growth Medium (Mouse) + Gentamicin per well, and medium was replenished every other day. Cultures were maintained in a 37 °C humidified atmosphere under 5% CO$_2$ for 10–12 days. Mammary organoids were maintained in culture by passaging them every 10–12 days at a 1:2 ratio split: Mammary organoids were washed by PBS and mechanically released from the Matrigel by breaking the matrix with a P1000 pipette and treated with TrypLE for 2–3 min at 37 °C to generate fragments. Following washes with trypsin inhibitor, mammary fragments were centrifuged at $100 \times g$ for 5 min at 4 °C. Supernatant containing single cells was removed and the fragments were resuspended in Matrigel, seeded in 24-well plates, and exposed to the described culture conditions.

**Organoid cryopreservation.** At times, organoids were grown for the complete 10–12 days, before removal from Matrigel by breaking the matrix with ice-cold DMEM followed by a PBS wash. Organoids were resuspended in freezing media (Cryostor 10, Sigma) at a density equivalent to two wells per ml freezing solution, and aliquoted into cryovials. Cryovials were stored overnight at −80 °C before long-term storage in liquid nitrogen. For thawing, vials were placed in placed in a 37 °C water bath and the contents washed twice in DMEM + 2% FBS (Invitrogen), before reseeding in Matrigel at the required density.

## CRISPR
**Cell preparation.** Mammary glands from 8–10-month-old virgin Brca2 Tr/WT and Brca2 WT/WT were processed to single cells as described above and seeded into six-well plates at a density of $1–2 \times 10^5$ cells/well. EpiCult™ Plus Medium (STEMCELL tech) + 50 mg ml$^{-1}$ Hydrocortisone (Sigma) + Gentamicin (Gibco) was added to the wells. Cultures were maintained in a 37 °C humidified atmosphere under 5% CO$_2$ for 8–10 days. Medium change every 3 days. Cells were trypsinized at 70–80% confluency, using pre-warmed 0.05% Trypsin at 37 °C for 5 min. The reaction was terminated by adding DMEM + 2% FBS. Cells were passed through a 40-μm cell strainer.

**Nucleofection for gene targeting.** In total, 1 μl Cas9 protein (5 μg/μl, Thermo Fisher) and 100 pM of synthetic guide RNA (Tspan8 guide sequence 5′-3′: GGGGAGTTCCGTTTACCCAA; Thrsp guide sequence 5′-3′: AGTCATGGATCGGTACTCCG; Merck) were mixed and incubated at RT for a minimum of 10 min to assemble the ribonucleoprotein

(RNP) complex. In total, $2 \times 10^5$ single cells were resuspended with Lonza P3 Primary Cell Nucleofector® Solution and mixed with the pre-formed RNP complexes and transferred to a Nucleocuvette™ (Lonza). Nucleofection was performed using programme EO-115 on the 4D Nucleofector™ X unit (Lonza). After nucleofection, the cells were immediately transferred back to warm EpiCult™ Plus complete medium to continue culture for a further 5–7 days.

**Genotyping.** Cells were harvested 5–7 days after nucleofection, and genomic DNA was extracted with the DNeasy Blood & Tissue Kit (Qiagen). Targeted regions were PCR amplified using Q5® High-Fidelity DNA Polymerase (NEB), with the corresponding primers listed. Tpan8 forward primer: AAGACACATCTCCGTAACGACA, reverse primer: AGCTCCCCTGGTGCTTACTG; Thrsp forward primer: CGGACTCT-GAGGAAGGAAGC, reverse primer: GGTGGAACTGGGCTTCTAGG. The products were gel purified using QIAquick PCR Purification Kit (Qiagen) and sent for Sanger sequencing.

## Immunofluorescent staining
Whole mounts: For all whole-mount immunofluorescence experiments, 4–5 Matrigel domes of organoids from each sample was pooled to ensure robust number of organoids were counted. Organoids were washed with 1× PBS and mixed with 1 mL of Cell Recovery Solution. The mixture was incubated on ice for 40 min, centrifuged at 300×g, then washed twice with 1× PBS. Organoids were fixed with 4% paraformaldehyde for 45 min at RT, washed 3× with PBS, 0.1% BSA, 0.2% Triton X-100 and 0.1% Tween 20 (Immunofluorescence (IF) Buffer) and then heat-induced antigen retrieval in citrate buffer (pH 6) for 20 min. Organoids were then permeabilised with 1% Triton X-100 in PBS for 1 h and blocked with 5% goat serum in IF buffer for 1 h. Organoids were incubated on a shaker for 24–48 h at RT with primary antibodies in IF buffer (Supplementary Table 1). After incubation with the primary antibodies, organoids were washed three times with IF buffer for 5 min and incubated with the secondary antibodies diluted in IF buffer Supplemented with 10% goat serum (Supplementary Table 2) and incubated on a shaker for 24–48 h. Nuclei were counterstained with DAPI. Organoids were cleared using fructose glycerol[54].

Organoids were imaged using the Zeiss 880 confocal microscope (Zeiss) or an Operetta CLS high content imaging system (PerkinElmer). An automated Preciscan analysis with Z method was established. Firstly, the entire well area was imaged with a deep Z-stack, in the DAPI channel only, at low resolution (×10 air objective) to locate the organoids. An analysis sequence was run over this first round of images to segment the organoids, and automatically pass their locations to a second round of higher-resolution imaging using the Harmony software's 'PreciScan' function. Each organoid was then rescanned with a stack of 16μm Z-step size using a 20×/1.0 water-immersion objective, positioned to the correct Z height, with three channels (DAPI, Alexa488, Cy3). The segmentations of both prescan and rescan images used the intensity of the DAPI signal, a pixel classifier, and texture information from filtering of the image, to create a set of consensus objects with in-focus DAPI signal.

Tissue sections: Intact mammary glands were freshly isolated and fixed in 10% neutral buffered formalin overnight before processing the tissue into paraffin. Tissue blocks were sectioned at 4 μm, deparaffinized and before performing heat-induced antigen retrieval in citrate buffer (pH 6). The samples were preblocked in PBS with 1% BSA and 0.1% Tween 20, then incubated with primary antibodies overnight at 4 °C (Supplementary Table 1). The secondary antibodies were goat anti-AF488, goat anti-Rabbit Cy3, and goat anti-AF647 (Supplementary Table 2; Jackson ImmunoResearch) and were all used at 2 μg ml$^{-1}$. A no primary antibody was used as a control. Slides were stained with DAPI to visualise the nuclei and sections mounted with ProLong Gold antifade (Invitrogen). Tissue sections were imaged using the Zeiss 880 confocal microscope (Zeiss) or the AxioScan microscope (Zeiss).

Analysis for tissue sections: Tissues images were collected from a confocal using the tissue stitching tool to recreate the tissue section or from whole tissue images scanned on an AxioScan microscope. Using FIJI/ImageJ (version 1.53q), regions were drawn around all glandular/ductal structures, using DAPI as a counterstain. Mean fluorescent intensity (MFI) was normalised to the region area to account for differences in gland sizes and the multiple regions were averaged for each tissue sample.

## Mass cytometry sample preparation and analysis

Mass cytometry and subsequent data analyses were performed as previously reported[20,28]. In this study, samples were barcoded using the Cell-ID 20-Plex Pd Barcoding Kit (Fluidigm) following the manufacturer's instructions. Mammary epithelial cells were collected from third and fourth mammary glands of virgin mice. Lymph nodes were removed, and glands minced with surgical scissors before enzymatical dissociation for 1.5 h in DMEM/F12 (1:1) Supplemented with 2 mg mL$^{-1}$ collagenase (Roche) + Gentamicin (Gibco). Samples were briefly vortexed every 30 min. Mammary gland fragments were treated with $NH_4CL$ to lyse red blood cells, then dissected to single cells with 0.05% Trypsin-EDTA (STEMCELL Technologies) and 5 mg ml$^{-1}$ disease (STEMCELL Technologies) and 1 mg ml$^{-1}$ DNAse (Sigma) and filtered through a 40-μm cell strainer (Falcon). Single-cell suspensions were adjusted to 5 million per mL cell concentration of cells. 0.5 μL cisplatin (Fluidigm) per 5 million cells were added to each sample and incubated for 5 min. Cells were then washed, and fixed with 16% and incubated 10 min. Following fixations cells were snap frozen in liqoud nitrogen and stored. For antibody staining: Samples were then pooled, resuspended in HBSS (Gibco) with 100 μg/mL DNAse I (STEMCELL Technologies) for 15 min in a 37 °C water bath, and washed once with Cell-Staining Medium (CSM; Fluidigm). Next, the pooled sample was stained at room temperature for 30 min with an antibody cocktail (Supplementary Table 4) containing the appropriate volume of each antibody with CSM. The sample was washed two times in CSM, fixed for 30 min at room temperature in 4% PFA (EMS) in PBS, and washed once with CSM. The sample was then suspended in 0.5 mL Intercalator Solution (300 μL CSM with 50 μL 16% PFA, 50 μL Fix-Perm Buffer (Fluidigm), and 0.67 μL Cell-ID Intercalator-Ir (Fluidigm)) per 1 million cells overnight at 4 °C. The sample was washed once the following day with CSM and twice with Milli-Q water. The sample was then resuspended in bead water (1:10 4-Element EQ Beads (Fluidigm) in Milli-Q water) to a concentration of ~750,000 cells/mL solution. Samples were run using the Helios Mass Cytometer.

CyTOF data were normalised to the bead signal, converted to an FCS format, and debarcoded using the CyTOF Software v7 from Fluidigm. Samples were gated in FlowJo for quality control parameters to yield live, single cells: Event Length, Gaussian Parameters, 140Ce-Beads, DNA content 191Ir, DNA content 193Ir, and Viability 195Pt (Supplementary Fig. 11). All single cells were then imported into the toolkit Scanpy[55], embedded as a UMAP, and clustered using the Leiden algorithm[56]. Clusters were then annotated as either HR- luminal, HR+ luminal, basal, immune, fibroblast, or vascular (Supplementary Fig. 12). Epithelial cells were isolated in silico and utilised for downstream analyses. Heatmaps of the CyTOF data were made in Scanpy using the pl.matrixplot function. Differential marker expression analyses were performed using the Scanpy function tl.rank_genes_groups function with default settings using the mean expression data for each sample in each epithelial cell type.

## Flow cytometry

Mammary glands: Dissociated to single cells and cells were then incubated with the following primary antibodies (Supplementary Table 3): CD31, CD45, Ter119, EpCAM, CD49f,CD49b and Sca1. Biotin-conjugated antibodies were detected with Streptavidin-eFluor450 (eBioscience). Cells were then filtered through a 30-μm cell strainer

(Partec) and incubated with DAPI and were sorted on a FACSAria II (Supplementary Fig. 13; Becton Dickinson). Organoids: 1–2 Matrigel domes were dissociated to single cells using TrypLE at 30 °C for 10–15 min, and triturating every 5 min. Single mammary cells were then incubated with EpCAM, CD49f, CD49b, Sca1 (Supplementary Fig. 14), and where required Tspan8 (Supplementary Figs. 15 and 16). Cells were then filtered through a 30-μm cell strainer and incubated with DAPI and were analysed by using an FACS Fortessa (BD Biosciences) (Becton Dickinson). Flow cytometry data were analysed using FlowJo (version 10. Tree StarInc). For BrdU-staining: BrdU (100 μM) was administered to organoids for 12 h then digested to single cells as described above. For intracellular staining, cells were first stained with the indicated surface markers and then fixed with BD Cytofix/Cytoperm Buffer (BD Bioscience) for 20 min at 4 °C, followed by incubation with BD Cytoperm Plus Buffer for 10 min at 4 °C and re-fixed for 5 min at 4 °C. Cells were then treated with DNase I (1 mg/ml, Sigma) in PBS and then immunostained with BrdU-FITC (Supplementary Fig. 17). For γ-H2AX staining: cells were stained with Zombie UV (BioLegend) instead of DAPI and then fixed with BD Cytofix/Cytoperm Buffer (BD Bioscience) for 20 min at 4 °C, followed by incubation with BD Cytoperm Plus Buffer for 10 min at 4 °C and re-fixed for 5 min at 4 °C. Cells were immunostained with Phospho-Histone H2A.X (Ser139)-FITC on ice for 30 min (Supplementary Fig. 18).

## Single-cell RNA sequencing

**Cell preparation.** For single-cell RNA-seq: cryopreserved organoids from P0, and P4 were thawed, reseeded into Matrigel, and cultured for 3 days. Organoids were dissociated into single-cell suspension as described above. Viability of <90% was confirmed on all samples and cells were immediately submitted to the CRUK CI genomics core facility for library preparation.

**Library preparation and sequencing.** Library preparation was performed according to instructions in the 10× Chromium single-cell kit version 3. The libraries were then pooled and sequenced across eight lanes on a NovaSeq6000 S2.

**Bioinformatics analysis of scRNA-seq data.** Cell Ranger 6.0.2 (http://10xgenomics.com) was used to process Chromium scRNA-seq output and generate the count table. Samples were demultiplexed using barcode assignment and unique molecular identifier (UMI) quantification. FASTQ reads were aligned to the mouse reference genome refdata-cell ranger-mm10-3.0.0. Seurat (V4.0) in R (V4.1.3) was used to carry out all analyses. Cells that met quality control conditions (unique number of genes between 1000 and 6000 and <5% of genes are mitochondrial genes per cell) were included for downstream analysis. Thirty-four thousand two hundred cells with expression levels for 3000 genes passed quality control. Individual samples were integrated, data normalised and scaled with the Sctransform function. The dimensional reduction was performed by PCA. The FindNeighbors function of Seurat was used to construct the Shared Nearest Neighbor (SNN) Graph, based on unsupervised clustering performed with Seurat function FindClusters. For visualisation, the dimensionality was further reduced using Uniform Manifold Approximation and Projection (UMAP) method with Seurat function RunUMAP. Differentially expressed genes were determined using Wilcoxon rank-sum test with $P$ value <0.05. We determined the major epithelial cell subsets whereby differentially expressed genes (DEGs) of each cluster were identified using the FindMarkers function in Seurat, which returns the gene names, average log fold-change, and adjusted $P$ value for genes enriched in each cluster. We carefully reviewed top 50 DEGs for each cluster with special focus on well-studied mouse mammary epithelial markers. These were integrated to define cell types and cell transcriptomic states. Mammary primary epithelial cell signatures were curated from several previously published mammary scRNA-seq

datasets[34]. The gene signatures of basal, HR− luminal and HR+ luminal mammary primary cells from these datasets was then calculated between the clusters in our data with the AddModuleScore function in Seurat. Heatmaps of the HR− luminal cell clusters were plotted using the doHeatmap Seurat function using the top 20 genes from each luminal cluster. Gene Ontology enrichment analysis of DEGs across the clusters and conditions was performed using the DEenrichRPlot function in Seurat. The EnrichR databases used were the GO_Biological_Process_2021, KEGG_2019_Mouse and MSigDB_Hallmark_2020.

### Whole-genome sequencing

**Low-pass whole-genome sequencing.** Shallow whole-genome sequencing was performed at Novogene's Cambridge Sequencing Centre. Passage 0 and 4 organoids were dissociated to single-cell suspensions and DNA extracted using the QIAamp DNA Micro kit (Qiagen). The extracted genomic DNA was sheared, and the fragments were end-repaired, A-tailed and further ligated with Illumina adaptors. The fragments with adaptors were PCR amplified, size selected, and purified. The library was checked with Qubit and real-time PCR for quantification and bioanalyzer for size distribution detection. The 150 bp paired-end sequencing reaction was performed, resulting in an average genome coverage of 3× per sample.

**CNA analysis.** Raw reads were trimmed for sequencing adaptors with fastp using default parameters. Trimmed reads were then aligned to the reference genome using Burrows–Wheeler Aligner (BWA). Subsequent processing, including duplicate removal was performed using samtools and Picard. Alignment statistics and genome coverage metrics were extracted using GATK and Picard.

CNA calling and analysis were performed using QDNAseq package in R. Bin annotations for mouse genome build were obtained from QDNAseq.mm10 package. Segmentation analysis was performed using default parameters, and the resultant output files were summarised using R.

**RNA-seq analysis on TCGA tumours with BRCA2 mutations.** RNA sequencing data from breast cancer samples ($n = 1091$) were collected from The Cancer Genome Atlas Programme (TCGA) repository from the TCGA-BRCA projects[57]. BRCA2 mutation status were obtained using the cBioPortal[58], which contained four partially overlapping TCGA-BRCA datasets. Excluding BRCA2 variants of unknown significance, 41 BRCA2-mutant cancer samples were identified.

After the stratification of BRCA2-mutant tumours, RNA-seq analysis was performed. Raw RNA-seq counts for breast (TCGA-BRCA project) cancer samples were obtained from TCGA repository. Read counts were filtered to remove transcripts with zero counts across all samples, then were log10 transformed. This was followed by R package DESeq2 workflow[59] to perform differential gene expression analysis with criteria $P$ value < 0.05 and log2(Fold Change) > 0.5. Pathway enrichment analysis was performed with ClusterProfiler [Yu 2012]. The Hallmark gene set was used from Molecular Signature Database (MSigDB)[60].

**Statistics.** Data are presented as mean ± SD or mean ± SEM, as indicated in the figure legends. All data were obtained from at least two independent biological replicates, and $n$ represents the number of independent mice or samples used for the analysis. $N$ values are indicated in the figure legends. Statistical significance was determined with a two-sided Mann–Whitney test or $t$ test with Welch correction and reported from GraphPad Prism v9. Differences were significant when $P < 0.05$. Differences around $P < 0.05$ were listed in the figure legends.

### Reporting summary

Further information on research design is available in the Nature Portfolio Reporting Summary linked to this article.

## Data availability

The single-cell RNA-seq data generated in this study have been deposited in NCBI's Gene Expression Omnibus and are accessible through GEO Series accession number GSE214539. The CyTOF data generated in this study have been deposited in Mendeley Data [https://data.mendeley.com/datasets/3nm9wnbndc/1][61] GO terms libraries from Enrichr (https://maayanlab.cloud/Enrichr/#libraries), including GO_Biological_Process_2021; MSigDB_Hallmark_2020 and KEGG_2019_Mouse were used for the analysis. BRCA2 mutation status were collated from cBioPortal (https://www.cbioportal.org/), querying TCGA-BRCA datasets. Hallmark gene set was used from Molecular Signature Database (https://www.gsea-msigdb.org/gsea/msigdb). All other relevant data are provided within the article, Supplementary information files and source data file. Source data are provided with this paper.

## Code availability

All code associated with this manuscript has been uploaded to GitHub (https://github.com/ms2140/Brca2_mammary/ and is available online in Zenodo https://zenodo.org/badge/latestdoi/542765593)[62].

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

## Acknowledgements

We thank Dr. Ben Hall (UCL) and Dr. Harveer Dev (Department of Oncology, Cambridge) for critical reading of our manuscript. We thank the CIMR flow cytometry facility for assistance with cell sorting and the CRUK Cambridge Institute genomics core facility, in particular Katarzyna Kania with preparing the samples for single-cell RNA sequencing, and the DFCI Mass Cytometry Core, led by Nicole Paul and Eric Haas. This work was funded by Gray Foundation Team Science Award to J.S.B. and A.R.V., Medical Research Council (MRC) Programme grants MC_UU_12022/1 and MC_UU_12022/8 to A.R.V., and by the Krishnan-Ang Fellowship to M.S.

## Author contributions

M.S. and A.R.V. conceived this study. M.G.N., G.K.G., L.R.K., K.G., D.P., H.N. and M.S. performed experiments and analysed experimental data, specifically M.G.N. performed the CRISPR experiments and analysis, G.K.G. performed the Cytof experiments and analysis, L.R.K. and K.G. performed shallow whole genomic sequencing, D.P. managed the mouse colony and collected tissues, H.N. performed the IF high content imaging and analyses and M.S. performed the IF staining, flow cytometry, organoid cultures, drug treatments, scRNA sequencing and analyses. M.S., J.S.B. and A.R.V. interpreted the results. M.S. and A.R.V. wrote the paper. All authors revised the manuscript and have approved the final version.

## Competing interests

The authors declare no competing interests.
