## [Peer Review File · Nature Communications]

A transcriptional response to replication stress selectively expands a subset of BRCA2-mutant mammary epithelial cellsREVIEWER COMMENTS

Reviewer #1 (Remarks to the Author):

The authors utilized Brca2mut/WT mammary organoids to investigate how BRCA2 mutations may affect mammary epithelial subpopulations with the goal to provide molecular insights into association of BRCA2 mutations with the development of luminal-like breast cancers. Their data showed that BRCA2 heterozygous loss does not affect epithelial compositions in mammary glands. Surprisingly they found that long-term replication stress induced by hydroxy urea leads to preferential survival and expansion of Brca2mut/WT HR-luminal cells. scRNA-seq analysis further showed that increased levels of Tetraspanin-8 and Thrsp mRNA, and replication stress survival pathways including Type I interferon responses are found in Brca2mut/WT cells. These data suggested that in response to replication stress, Brca2 heterozygosity triggers the expression of interferon-responsive and mammary alveolar genes and thus enhances outgrowth of HR- luminal cells. The new findings from this manuscript are interesting and will help understand how BRCA2 loss is associated with luminal-like breast cancer development. Overall, the conclusion is supported by experimental data based on testing HU-induced replication stress in Brca2mut/WT mammary organoid models. However, there are several concerns regarding the experimental design and data interpretation.

1. Is this transcriptional response, interferon-responsive changes specific to Brca2-mutant cells or can this be found in Brca1mut/WT cells in the presence of HU? Would Brca1mut/WT cells favor outgrowth of luminal cells or basal cells in prolonged HU exposure since BRCA1 and BRCA2 share some common functions in regulating HU-induced replication stress response?
2. In Figure 2D, the BrdU labeling representative flow data does not match the quantitative bar graph. Brca2mut/WT cells showed lower number of BrdU positive cells in the presence of HU. It is not clear whether the increased BrdU labeled cells in Brca2mut/WT cells were due to increased BrdU uptake (increased proliferation/S phase cells) or impaired regulation of cell cycle progression (cells accumulated in S phase). BRCA2 is also required for proper replication stress signaling. It seems that gamma-H2AX percentage was not changed significantly (Figure 2B). p53 induction is the master regulator of DNA damage-induced cell cycle checkpoints. Is p53 induction in response to HU impaired in Brca2mut/WT?
3. As shown in Figure 2B, DNA damage seems to be relatively similar, what could be the mechanism underlying the activation of interferon-responsive genes in Brca2-mutant cells in the presence of HU treatment? In addition to outgrowth of organoids, does the activation of interferon-responsive signaling promote metastatic gene changes? Previous study indicated that chromosomal instability induced cGAS-STING-type I interferon response promotes breast cancer malignant transformation or metastatic phenotypes.
4. Estrogen has been recognized as a physiologically relevant replication stress-inducing stimulus. It would be more relevant to test whether estrogen treatment-induced replication stress may promote luminal subpopulation outgrowth in Brca2mut/WT cells compared to using HU as a replication stress stimulus.
5. In the TCGA breast tumors, can the increased levels of Tetraspanin-8 and Thrsp mRNA be found to correlate with BRCA2-mutant luminal tumors? A validation correlation in the TCGA tumors may help support the hypothesis.

Reviewer #2 (Remarks to the Author):

The manuscript entitled, "A transcriptional response to replication stress selectively expands a subset

of BRCA2-mutant mammary epithelial cells” seeks to identify the molecular mechanism underpinning the luminal predisposition of BRCA2 mutated breast cancers.

The authors rely upon a combination of organoid culture, scRNAseq, CyTOF and histological analyses to investigate this question. The authors start by showing that Brca2 mutant tissue and organoids from a mouse model are not at baseline grossly altered compared to (age and estrous matched) wildtypes. Then, considering the known role of BRCA2 in DNA repair and replication stress, the authors use hydroxyurea (HU) as a way to perturb Brca2 mutant cells to understand their unique growth response to this cellular stressor. Through characterization of organoids, mostly with scRNAseq and flow cytometry, after exposure to HU, the authors identify a Brca2-specific expansion of cycling HR- luminal progenitor cells, which they argue are defined by their enriched expression of Tspan8. A compelling finding of the paper is in Figure 6E in which the luminal Brca2 growth advantage is lost when Tspan8 is removed.

However, the strength of this finding would be augmented by studies using the Brca2 mutant mice (for example flow from mammary tissue) from which organoids were derived in order to demonstrate that these findings do not suffer from any artifacts specific to 3D culturing methods. For example, Brca2 organoid (or Tspan8+ sorted cells from organoids) transplantation followed by tumor tracking could be completed to validate the functional relevance of these findings.

Major concerns:

- A justification for why organoids, but not tissue from the mice, are used exclusively is required (ideally in the introduction)
- Because young and old mice are used at first, but not always, the rationale for this choice should be more thoroughly presented
- The organoids shown in figure 1F display limited branching and are mostly of the “cystic” phenotype. An investigation or explanation of this finding should be included
- The heatmap shown in figure 5B does not have any genes labelled, therefore it provides little value to the reader. Consider showing only part and labelling the genes with the heatmap enlarged, or consider removing this panel
- Ideally GSEA analysis would be completed alongside GO term analysis for validation (Figure 5C)
- References for the assertion that Tspan8 is a mammary gene (on line 271) should be added
- The experiment completed in Figure 6C should have at least n=3 samples per condition, ideally n=5
- Confirmation of the Tspan8 luminal population should be confirmed using flow on tissue from Brca2 KO mice, alongside organoids, to confirm this population is not merely specific to organoid culturing of Brca2 cells
- It would be helpful to complete an analysis of exposure of Brca2 organoids to a stressor that will induce CNA such that the negative finding in Figure 3C and Supplementary 3C can be confirmed by a positive control for the assay

Minor concerns:

- A definition of replication stress as it is considered by the authors would be useful to include in the introduction
- Have statistics been run for Fig 1C to confirm non-significance? If so, this would be great to mention in the figure legend
- On line 102, please clarify/define, “unchallenged”
- Is there a reason organoids were derived from single cells rather than clusters of cells? If so, please describe this further in the manuscript
- In line 135, is the 5-6 mice per condition of the age in weeks of the mice used?
- For the experiment performed for Figure 3, were there technical replicates or only biological replicates?
- It would be helpful to show a representative example for the gating strategy used to derive the MFI

data for Supplementary Figure 1A

- In figure 4B, it would be visually helpful to group cell types together, for example, not splitting up the luminal populations with the two basal populations in the middle
- Citations should be included to justify the use of certain markers for various populations in scRNAseq (paragraph starting on line 203)
- The origin of the gene lists used for the scores derived in Figure 3C need to be shared
- For figure 4E, please list the control conditions before the HU conditions
- AZD1480 should be clarified as a JAKi in the manuscript, rather than just assumed based on the figure label
- When it is stated on line 271 that cluster 4 is “defined by” high expression of Tspan8 and Thrsp, how was this determined? What was the log₂fc value for this gene in cluster 4 relative to all other clusters (via FindAllMarkers function)? Is this gene a top gene for the cluster or just one that was used for the DotPlot shown? These strategies were described clearly in the methods but could be stated in reference to this particular gene in the manuscript
- In figure 5, it would be helpful to show a proportion of cells graph as in Figure 4E to better appreciate the relationship of cluster 4 to the Brca2 genotype and the HU treatment
- As all experiments are completed using mouse cells, the title should be edited such that Brca2 is written not in all caps
- What is the meaning of neutrophil genes displayed as upregulated in epithelial cells (Supplementary 5B, D)? Might this relate to exposure to reactive oxygen species?

Reviewer #3 (Remarks to the Author):

In this manuscript, Najafabadi and colleagues explore the phenotype of BRCA2 heterozygous organoids expressing a pathogenic mutation and address the question of whether BRCA2 mutation per se affects the fate of mammary epithelial cell subpopulations using murine organoid models. Using acute versus chronic exposure to replication stress induced by HU and analyzing subpopulations by scRNAseq the authors find that Brca2mut/wt cells activate a transcriptional response to chronic replication stress that favors the outgrowth of hormone receptor-negative luminal cells. Moreover, they identify genes of the interferon response and mammary alveolar genes as responsible for this phenotype.

This is a very difficult and important question to address; indeed, BRCA2 mutated tumors are very different from BRCA1 mutated ones in that BRCA2 mutated tumors are generally luminal-like and HR+. This is very different from BRCA1-mutated tumors which are generally basal-like and triple-negative. The reason for this is still ill-defined.

Intriguingly, the authors find expanded mammary luminal populations that are HR- in Brca2mut/wt organoids just like what it has been found in BRCA1-mutated tumors; however, they do not comment on why or how do they explain that HR-luminal cells expand in their model whereas BRCA2-mutated tumors are generally HR+. So then, what is the relationship between BRCA2 tumorigenesis in mice versus in humans? This manuscript falls short on answering this underlying question and needs to be addressed to determine whether their model recapitulates or not what happens in human BRCA2-mutated tumors and overall, what are the clinical implications of this work.

Also, I wondered why they chose breast tissue from aged versus young given that BRCA2-mutated breast cancer occurs at early onset as compared to the general population.

There are some other major points that have to be addressed to consider publication as well besides these general ones that are detailed below:

Fig. 1D. The authors state that Het-aged mice display an increase in HR-luminal markers. However, in the graph, we can see that the Het-young mice display an increase, not the old ones, where the

expression of those markers is even reduced compared to Wt-young.

Fig. S1B. In this figure, the gene expression does not correlate with the one shown in the heat map (fig 1D): in fact in S1B, the Het-aged mice display an increased expression of the genes CD14 and CD36. However, when looking at the expression of the same genes in fig. 1D we can see a strong expression (intense red box) in the Het-young mice whereas the Het-aged show a reduced level (white box). Could it be an error in the legend in Fig. 1D?

Fig.2D: how come HU-treated cells (both WT and Brca2-mutated) proliferate more than their not-treated counterpart? HU is known to slow replication so we expect the opposite trend. Moreover, the levels of BrdU+ cells differ from the ones in Fig5E under the same conditions.

Fig. 3B: These results seem not conclusive as the error bars are quite big between -HU + HU in the mut/wt cells. As this is an important result, a complementary assay such as the FACS showing the number of BrdU+ cells in the two conditions as shown in Fig. 2 would be more convincing.

Also, are these cells HR-luminal? Please specify in the legend.

Fig 3D, S3D and S3E: The authors state that in figure S3D the multi-lineage organoids – corresponding to the black box, were present in 60, 33, 33 and 50 of Wt, Wt-Hu, B2 mut and B2 mut+HU; these numbers do not seem to correspond with the bar graphs. Please correct.

Fig. 3F: There is quite some variability between different organoid cultures in Brca2 mut/wt which leads to a difficult assessment of the changes. It would be much more informative/convincing to show the paired organoids (HU-treated vs untreated) to determine whether the trend is similar in all or not and be able to conclude.

Fig. 5G: the graph shows that both WT cells and Brca2mut/wt HU-treated show an increase in Tspan8+ cells when treated with IFN β compared to untreated cells however there is no comment on that in the text. What is the significant relevance of this if the difference is already there in the non-HU WT condition? In general, this Fig. and the interpretation of the data are unclear. For example, Jaki is supposed to affect organoid size independent of proliferation but later on, is stated that it affects proliferation. Please reanalyze the data and clarify the conclusions. Then change accordingly the statements in lines 337-339.

Fig. 6C. In the text they refer to this panel stating that Tspan8 expanded in the HR-luminal cells but there are no stats on the graph (or they are not significant) so please clarify this statement/graph.

Fig. S5.A: In the WT-C organoids there is a similar proportion of cells belonging to cluster 4, similar to B2-HU. Moreover, following HU, in the WT there is an expansion of cluster 5, is it also associated with interferon response?

Fig. S5.B: cluster 1 of Brca2-HU treated cells is associated with IFN response, how about cluster 1 of WT-C, which is even more present than in B2-HU?

Minor

Introduction: When referring to Ref. 6 the authors state that there are no phenotypic changes in myoepithelial cells in Brca2 mutated tumors whereas the ref. states the opposite. Please correct.

Fig. 3B. Het control should be labeled in light blue to follow the same color logic as in the other graphs and avoid confusion.

Fig. 3E. The authors state that there is a significant expansion in the HR- compartment although in the figure we can only see that the Het-HU organoids display an increase in the CD49+ but no in the DN compartment, where there is no difference across the conditions.

Fig 4D: in the text, the authors state that the expression of the luminal alveolar gene Aqp5 is elevated in the B2_HU_P4, however, when looking at the figure we can see that for this gene there is no difference in expression when compared to WT P0 or P4.

Fig. 5F: The interpretation that Jaki reduced organoid size only in the HU-treated Brca2- mutated organoids seem to be an overestimation considering the variability in the replicas in this condition; is it not significant the reduction observed in the other groups upon AZD1480 treatment?

Fig. S3A: The authors conclude from this figure that all the epithelial sub-populations display similar levels of DNA damage across the different groups; however, they fail to comment on why there are

different levels of damage in the different lineages, in particular why the treated WT accumulate more gH2AX in the basal and HR+ compartment.

REVIEWER COMMENTS

Reviewer #1 (Remarks to the Author):

The authors utilized *Brca2*mut/WT mammary organoids to investigate how BRCA2 mutations may affect mammary epithelial subpopulations with the goal to provide molecular insights into association of BRCA2 mutations with the development of luminal-like breast cancers. Their data showed that BRCA2 heterozygous loss does not affect epithelial compositions in mammary glands. Surprisingly they found that long-term replication stress induced by hydroxy urea leads to preferential survival and expansion of *Brca2*mut/WT HR-luminal cells. scRNA-seq analysis further showed that increased levels of Tetraspanin-8 and *Thrsp* mRNA, and replication stress survival pathways including Type I interferon responses are found in *Brca2*mut/WT cells. These data suggested that in response to replication stress, *Brca2* heterozygosity triggers the expression of interferon-responsive and mammary alveolar genes and thus enhances outgrowth of HR- luminal cells. The new findings from this manuscript are interesting and will help understand how BRCA2 loss is associated with luminal-like breast cancer development. Overall, the conclusion is supported by experimental data based on testing HU-induced replication stress in *Brca2*mut/WT mammary organoid models. However, there are several concerns regarding the experimental design and data interpretation.

We thank the Reviewer for their positive remarks. Below, we provide a point-by-point rebuttal including substantial new experimental data to address the concerns raised.

1. Is this transcriptional response, interferon-responsive changes specific to *Brca2*-mutant cells or can this be found in *Brca1*mut/WT cells in the presence of HU? Would *Brca1*mut/WT cells favor outgrowth of luminal cells or basal cells in prolonged HU exposure since BRCA1 and BRCA2 share some common functions in regulating HU-induced replication stress response?

We thank the Reviewer for raising this interesting point. *Brca1* mutant murine models are not available in our laboratory, precluding the investigation of differences between *Brca1* versus *Brca2* mutant mammary cells in their response to HU. We cannot therefore address whether IFN response gene activation after HU exposure is unique to *Brca2*-mutant mammary organoids. However, there are several reports indicating specific effects of replication stress on luminal progenitor cells. For example, luminal progenitor expansion has been reported in human *BRCA1*-mutant mammary glands [doi: 10.1038/nm.2000, doi: 10.1083/jcb.201804042], possibly because luminal progenitor programs are activated in response to environmental induction of DNA damage. In addition, mouse *Brca1*-deficient mammary cells display a similar hormone-independent expansion of luminal progenitor cells that is linked to the replication-associated DNA damage response, where proliferation of mammary progenitors is perpetuated by damage-induced, autologous NF-κB signalling [doi.org/10.1016/j.stem.2016.05.003].

We have now revised the Discussion (Page 11, lines 429-437) to summarize these issues.

2. In Figure 2D, the BrdU labeling representative flow data does not match the quantitative bar graph. *Brca2*mut/WT cells showed lower number of BrdU positive cells in the presence of HU. It is not clear whether the increased BrdU labeled cells in *Brca2*mut/WT cells were due to increased BrdU uptake (increased proliferation/S phase cells) or impaired regulation of cell cycle progression (cells accumulated in S phase). BRCA2 is also required for proper replication stress signaling. It seems that gamma-H2AX percentage was not changed significantly (Figure 2B). p53 induction is the master

regulator of DNA damage-induced cell cycle checkpoints. Is p53 induction in response to HU impaired in Brca2^{mut}/WT?

Concerning the flow cytometry data:

First, we do not believe there is any inconsistency between the representative flow data in the flow pseudo dot plot and the bar graphs shown in Figure 2D. The pseudo dot plots represent a single experiment, whereas the bar graphs show individual data points from several different experiments (including the representative example in the pseudo dot plot). In addition, we emphasize that the pseudo dot plot shows changes in the percentage of BrdU+ HR- luminal cell population alone, whereas the bar graph plots changes in several different epithelial subpopulations. Nevertheless, we have now included new data and additional samples (n=5) to strengthen our conclusions [updated Fig 2d]. Notably, we still observe a significant increase in the percentage of BrdU+ cells in the HR- luminal population at the passage 0 time point following HU exposure of both wildtype or Brca2^{mut}/WT organoids.

Second, the Reviewer suggests that there may be an increase in BrdU+ cells in Brca2^{mut}/WT samples after HU exposure, and enquires about connections with cell cycle progression. Accordingly, we have undertaken statistical comparisons of the BrdU+ cells between different epithelial subpopulations in the HU treated wildtype and Brca2^{mut}/WT organoids (ANOVA multiple comparison test: basal – p=0.0411, CD49b+ luminal – p= 0.1940, DN luminal – p= 0.0677, Sca1+ luminal – p=0.2137). Thus, the percentage of BrdU+ cells is not significantly different between the WT and Brca2^{mut}/WT in the luminal subpopulations, although there is a modest difference in the basal subpopulation alone. We have shown a more representative flow pseudo dot plot that reflects these observations in updated figure 2D. Overall, our results suggest that the luminal subpopulations in both WT and Brca2^{mut}/WT organoids exhibit similar BrdU incorporation after HU exposure.

We have now revised these points in the Results section. Page 4 lines 155-160, Fig 2d and New supp Fig 3b.

Concerning p53:

The Reviewer is correct to note that replication stress typically induces p53 activation, whereas we do not observe detectable p53 staining in Brca2 het organoids tested 24 hours after HU exposure.

To further investigate this issue, we first verified that the DO-1 antibody we could detect p53 expression in (control) MCF10A cells treated with 10uM etoposide for 4 hours. As shown in the Western blot provided below, DO-1 detected a band of the appropriate size in this control.

We also tested in further detail whether p53 could be detected by Western blotting in organoids from several WT and Brca2^{mut}/WT samples at passage 0 and passage 4. We treated these organoids as follows: C, untreated; E2, 1 uM estradiol for 24 hours; HU, 1mM HU for 24 hours. Cell extracts were prepared immediately following the treatments, before Western blotting with an anti-p53. As shown in the western blot below, we did not detect p53, apart from a faint band in two of the Het control samples at passage 0.

There are several possible explanations for this result. One possibility is that the p53 response to HU may occur over a short timeframe in primary organoid cultures. For example, earlier work suggests that p53-mediated p21 gene expression peaks within 10 h after irradiation of mammary organoids [doi.org/10.1016/j.celrep.2019.06.043], whereas our experiments were performed over a 24 h timeframe. Nevertheless, because we do not observe p53 induction **in either WT or Brca2 het organoids** 24 hours after HU exposure, these results are insufficient to support the Reviewer's

suggestion that Brca2 het mammary epithelial cells are defective in their response to HU. We have revised the text to explain these points.

A) Passage 0 or B) Passage 4 organoids were treated as follows: C – untreated; E2 – 1µM Estradiol for 24 hours; HU – 1mM HU for 24 hours. Western blot with anti-Brca2 (upper), HSP90 (loading control, middle) and anti-p53 (lower) antibodies. MCF10a cells treated with 10µM Etoposide for 4 hours were used as a control for the p53 antibody. WT = wildtype; Het = Brca2^{mut/WT}

3. As shown in Figure 2B, DNA damage seems to be relatively similar, what could be the mechanism underlying the activation of interferon-responsive genes in Brca2-mutant cells in the presence of HU treatment? In addition to outgrowth of organoids, does the activation of interferon-responsive signaling promote metastatic gene changes? Previous study indicated that chromosomal instability induced cGAS-STING-type I interferon response promotes breast cancer malignant transformation or metastatic phenotypes.

We performed further gene enrichment analysis including KEGG and MSigBD analyses to determine whether Brca2^{mut/WT} cells treated with HU incurred expression of metastatic or malignant transformative genes [New Supp Fig 6]. GSEA analysis revealed viral and interferon pathways elevated in Brca2^{mut/WT} HU treated cluster 4 as we reported in Figure 5C. We also observed an enrichment of a EMT pathway within cluster 3, including the following genes [LGALS1; GADD45B; TPM2; TPM1; HTRA1; CAPG; PTX3; TIMP1; THBS1]. We further investigated the cGAS-Sting pathways, using genes detailed in the Cytosolic DNA-sensing pathway from mmu04623 KEGG and observed an overall increase in several key genes within this pathway in the passage 4 organoids compared to passage 0. However, when comparing Brca2^{mut/WT} and WT organoids, there were no significant differences suggesting that cGAS-Sting pathway wasn't exclusively activated in Brca2^{mut/WT} treated organoids [new supp fig 6]. We have now revised the manuscript to discuss these points in the Results section. Page 8 lines 295-303, New Supp Fig 6e-g.

4. Estrogen has been recognized as a physiologically relevant replication stress-inducing stimulus. It would be more relevant to test whether estrogen treatment-induced replication stress may promote luminal subpopulation outgrowth in Brca2^{mut/WT} cells compared to using HU as a replication stress stimulus.

We thank the reviewer for an excellent suggestion. We repeated the longitudinal workflows with estradiol (E2) and also performed HU treatments alongside as a comparator using 5 biological independent replicates for WT and Brca2^{mut/WT} samples. Initially we observed that relevant physiological levels of E2 for cell culture (10nM or 100nM) did not induce an increase in DNA damage compared to untreated, and as such, we increased the amount of E2 administered to 1µM. This concentration was used by Zhang and colleagues [10.1016/j.cellsig.2016.08.001].

Wildtype organoids treated with 1nM or 100nM E2 for 24 (grey) or 48 (red) hours and assessed for γ H2AX+ cells (upper panels) or BrdU+ cells (lower panels) in the epithelial subpopulations.

Unlike HU-treated samples which displayed decreased organoid numbers at passage 4 [New Fig3b], E2 treated organoids remained consistent in organoid number alongside control WT and $Brca2^{mut/WT}$ samples [New Supp Fig 4]. We did observe a significant decrease in basal proportions and an increase in HR- luminal cell proportions especially at passage 4 [New Supp Fig 4]. However, these observations were consistent in both WT and $Brca2^{mut/WT}$ samples. We further tested whether there was an increase in BrdU incorporation following E2 treatment, and again, no discernible differences were observed in any of the epithelial subpopulations [New Supp Fig 3]. We next examined whether the increase in Tspan8+ cells in the HR- luminal population was only observed following HU treatments. We did observe an increase in Tspan8 + cells in the HR- luminal compartment for both WT and $Brca2^{mut/WT}$ E2 treated organoids [New Supp Fig 8a], again demonstrating that E2 treatment does promote outgrowth of the HR- luminal compartment, however it is not specific to $Brca2$ mutant genotype. Further to this, we next investigated whether E2-treated organoids influences the interferon response. We repeated the Jaki and INFb treatments with the E2-treated organoids. We observed that Jaki reduced growth in both controls and E2-treated organoids [New Supp Fig 7]. However, treatment with INFb had no effect on growth inhibition of E2 treated organoids [New Supp Fig 7].

These results indicate that although E2 treatment did induce an expansion in HR- luminal cells, this expansion was not specific to $Brca2^{mut/WT}$ cells.

We have now included these findings in the Results sections, pages 5, lines 166-168, 178-179, 191-193; page 6, lines 219-220, 227-230, 236; page 8, lines 319-320, 327-329, 335-340; page 9, lines 353-356, Figure 3b and Supplemental Figures 3a-c, 4a, 4d-j, 7a-e, 8a.

5. In the TCGA breast tumors, can the increased levels of Tetraspanin-8 and Thrsp mRNA be found to correlate with BRCA2-mutant luminal tumors? A validation correlation in the TCGA tumors may help support the hypothesis.

We have investigated *TSPAN8* and *THRSP* in the TCGA breast cancer dataset. *BRCA2* mutation status was obtained using cBioPortal. The data indicate that in both *BRCA2*-mutant and non-mutant cancers, both *TSPAN8* and *THRSP* expression is decreased. Thus, TCGA data differ from our findings on mouse-derived organoids, where *Tspan8* mRNA was increased, alongside an increase in HR-

luminal cells, in the *Brca2*^{mut/WT} organoids. We surmise that *Tspan8* upregulation could be an early response that is not sustained during tumour progression.

We next investigated the interferon responses in BRCA2 breast cancers. We combined germline and somatic *BRCA2* mutations to increase the number of samples. We investigated the pathways that are upregulated in *BRCA2*-mutant tumours and discovered that *BRCA2*-mutant tumours had upregulated interferon pathway genes (new Supp Fig 9c). This result parallels our data, in that *Brca2* mutant organoids exposed to DNA damaging agents also increased transcripts of interferon response related genes. Several studies using HR- luminal or triple negative breast cancer cells deficient in *Brca2/BRCA2* demonstrated activation of an immune or interferon response [doi.org/10.1038/s41467-019-11048-5, doi: 10.1002/path.4404, doi: 10.1038/s43018-020-00139-8]. Thus, an interferon response may be part of the HR- luminal population survival advantage, which would be a possible mechanism directing towards a triple negative subtype.

We have now revised the manuscript in the results section on page 10, lines 393-405. Discussion section page 12, lines 459-462, Supplemental Figure 9a-c.

Reviewer #2 (Remarks to the Author):

The manuscript entitled, “A transcriptional response to replication stress selectively expands a subset of BRCA2-mutant mammary epithelial cells” seeks to identify the molecular mechanism underpinning the luminal predisposition of BRCA2 mutated breast cancers.

The authors rely upon a combination of organoid culture, scRNAseq, CyTOF and histological analyses to investigate this question. The authors start by showing that *Brca2* mutant tissue and organoids from a mouse model are not at baseline grossly altered compared to (age and estrous matched) wildtypes. Then, considering the known role of BRCA2 in DNA repair and replication stress, the authors use hydroxurea (HU) as a way to perturb *Brca2* mutant cells to understand their unique growth response to this cellular stressor. Through characterization of organoids, mostly with scRNAseq and flow cytometry, after exposure to HU, the authors identify a *Brca2*-specific expansion of cycling HR- luminal progenitor cells, which they argue are defined by their enriched expression of *Tspan8*. A compelling finding of the paper is in Figure 6E in which the luminal *Brca2* growth advantage is lost when *Tspan8* is removed.

We thank the Reviewer for their positive remarks.

However, the strength of this finding would be augmented by studies using the *Brca2* mutant mice (for example flow from mammary tissue) from which organoids were derived in order to demonstrate that these findings do not suffer from any artifacts specific to 3D culturing methods. For example, *Brca2* organoid (or *Tspan8*+ sorted cells from organoids) transplantation followed by tumor tracking could be completed to validate the functional relevance of these findings.

We agree with the Reviewer that *in vivo* experiments using murine models could provide supporting evidence for our *in vitro* findings, but believe that the suggested experiments fall beyond the scope of this paper for several reasons. First, methods that have been developed in Brisken (doi: 10.1016/j.ccell.2016.02.002) lab to transplant organoids into murine mammary nipples to observe tumour outgrowth typically take many months to complete, and exhibit a low frequency of tumour growth that would require careful optimization of injection conditions for each genotype. Second, *Brca2* het mice do not typically develop spontaneous mammary tumours (PMID: 9537225), raising the possibility that *Brca2* het organoids will not progress to tumorigenesis after nipple

transplantation. Given the questionable nature of the possible outcomes, and the time and resource required for completion, we believe that such *in vivo* experiments would not be feasible for this paper.

Major concerns:

- A justification for why organoids, but not tissue from the mice, are used exclusively is required (ideally in the introduction)

We have now expanded the introduction to include justification of why organoids were used for this study.

“Organoids have enabled examination of tissue physiology, especially over time, which would not have been possible in an *in vivo* setting. Mammary organoids have been shown to faithfully maintain the phenotypes of all three epithelial subtypes and are an ideal model to mimic *in vivo* behaviours. Primary tissue samples can be multiplexed in a manner that allows control and treatments to be performed, reducing the variability observed when using *in vivo* samples. Organoids can expand the throughput of the experimental platform, without incurring the high costs and time associated with *in vivo* animal studies. Furthermore, organoids are more readily genetically manipulated than mouse mammary glands.”

Page 2 lines 69-75.

- Because young and old mice are used at first, but not always, the rationale for this choice should be more thoroughly presented

Because the incidence of breast cancer in female *BRCA2* mutation carriers increases at perimenopausal age, we wanted to examine mammary glands that would be comparable to a perimenopausal female. Adult mammary glands in mice are generally considered to form from 12+ weeks of age, which is equivalent to a young adult female, and mammary glands from an aged murine cohort are predicted to be equivalent to a mid-aged human female [doi.org/10.1016/B978-012369454-6/50074-1]. We utilised young and aged mammary glands to eliminate possible epithelial differentiation changes. Since we confirmed that epithelial proportions and cell types were unchanged, we decided to work on the aged cohort, as the mammary glands would have been exposed to more hormonal cycles than younger mice, again mimicking scenarios similar to the increase in hormone cycle exposure in female human breast. This point is now explained in the text. Updated results page 3, lines 112-118.

- The organoids shown in figure 1F display limited branching and are mostly of the “cystic” phenotype. An investigation or explanation of this finding should be included

We have now included a more detailed explanation of the sphere/cystic-like phenotype observed in the HR+ luminal cell type.

In vitro culture assays and *in vivo* lineage tracing experiments have reported that the HR+ luminal population to be lineage restricted and unable to form complex mammary branched structures. Our organoid system confirms the lineage restricted growth ability of HR+ luminal cells, which form only sphere-like phenotypes, consistent with previous reports [doi: 10.1186/BCR3334, doi: 10.1016/J.CELREP.2017.02.071, doi: 10.1038/S41467-020-15548-7].

In results section page 4 lines 136-138.

- The heatmap shown in figure 5B does not have any genes labelled, therefore it provides little value

to the reader. Consider showing only part and labelling the genes with the heatmap enlarged, or consider removing this panel

We have now included several genes for each cluster and labelled them on the heatmap. Updated Fig 5b.

- Ideally GSEA analysis would be completed alongside GO term analysis for validation (Figure 5C)

We have performed analysis using the MSigDB and KEGG databases and have updated the results and discussion. Although pathway names are different, similar genes are detected in the identified pathways [Supp Dataset 3]. These new results further validate that an interferon response is being upregulated.

Results section: page 8, lines 295-308, Supplemental Figure 6e-g.

- References for the assertion that *Tspan8* is a mammary gene (on line 271) should be added

We have inserted the Fu et al., [doi: 10.1038/ncb3471] reference for this sentence.

- The experiment completed in Figure 6C should have at least n=3 samples per condition, ideally n=5

We have performed an additional n=5 and combined the data, which has been updated in new fig 6c. The additional samples show similar trends that *Tspan8* expression is increased in the *Brca2*^{mut/WT} HU treated samples, but surprisingly, no change was observed in the WT.

Updated Figure 6c.

- Confirmation of the *Tspan8* luminal population should be confirmed using flow on tissue from *Brca2* KO mice, alongside organoids, to confirm this population is not merely specific to organoid culturing of *Brca2* cells

We thank the reviewer for this suggestion, but they may not have noticed that we have already analysed *Tspan8* expression in tissue from WT and *Brca2* mutant mice. As shown in Figure 1D, our Cytof antibody panel used to analyse tissue from *Brca2*-mutant mice included *Tspan8* antibody. Data in Figure 1D confirms *Tspan8* expression in both our primary WT and *Brca2*^{mut/WT} mammary epithelial tissues where it is predominantly expressed in the HR- luminal populations.

- It would be helpful to complete an analysis of exposure of *Brca2* organoids to a stressor that will induce CNA such that the negative finding in Figure 3C and Supplementary 3C can be confirmed by a positive control for the assay

We appreciate the Reviewer's suggestion, and attempted an experiment to provide a positive control, using organoids treated with gamma irradiation, which is well known to induce chromosomal breakage and CNAs, as well as HU. Exposing organoids to 2Gy and 5Gy irradiation showed a marked increase in DNA damage, that still was elevated 6 hours post damage (n=1).

Organoids irradiated with 2Gy (dark grey) or 5Gy (light grey) were harvested immediately (0 hours) or 6 hours following irradiation and assessed for γ H2AX+ in the basal (left), HR- luminal (middle) and the HR+ luminal (right) populations. Wildtype – WT; Brca2^{mut/WT} – Het.

To allow for the possible expansion and selection of non-random CNAs, we exposed the organoids weekly to 4mM HU or 2Gy irradiation. Following 4 weeks of treatments, the organoids were collected, as the 4mM HU treated organoids, irrespective of genotype, were showing signs of growth regression. We performed low-pass 3x WGS as previously described in the manuscript. However, we were unable to detect any CNA in the 4mM HU or 2Gy irr exposed organoids compared to controls [n=2, see figure below]. In parallel, we included as a positive control for our CNA detection pipeline of two different murine tumour samples, which exhibited a significant number of CNAs.

We note that the detection of random CNAs in organoid systems using low-pass 3x sequencing is challenging, since any given CNA is likely to be very rare unless it is enriched by selection. It is therefore possible that in the 4 week timeframe of these experiments, non-random CNAs do not occur with sufficient frequency to be detected by our approach. We have revised the text to state this caveat to our negative findings.

Page 5 lines 201-203.

A) Bar chart of the total CNVs detected of sWGS in wildtype or Brca2^{mut/WT} organoids from control, 4mM HU or 2Gy irr treated organoids. Data presented as mean +/- SD (n=2). B) aCGH plots from wildtype (upper) or Brca2^{mut/WT} (lower) organoids of controls at day 0 or following weekly treatments of control, 4mM HU or 2Gy Irr for 4 weeks. CNAs were absent in all samples. Representative plots from 2 independent experiments are shown. C) aCGH plots from murine tumour samples (tumour 1 and 2) used as positive controls for the analysis pipelines.

Minor concerns:

- A definition of replication stress as it is considered by the authors would be useful to include in the introduction

We define replication stress as a challenge that increases the frequency of stalled DNA replication forks.

Page 1 lines 31-32.

- Have statistics been run for Fig 1C to confirm non-significance? If so, this would be great to mention in the figure legend

We performed an ANOVA statistical analysis test, and this is noted in the legend of Figure 1C. We report no statistical differences observed. Page 25, lines 903-906.

- On line 102, please clarify/define, “unchallenged”

We define unchallenged in this context as being in the absence of additional known tumour driver mutations or being exposed to genotoxins. We have now included this clarification on page 3 line 112-114.

- Is there a reason organoids were derived from single cells rather than clusters of cells? If so, please describe this further in the manuscript

We wanted a quantifiable way of measuring organoid numbers at the initial passage, and to have control of the amount of cellular material we were inputting into the organoid cultures. Dissociating the tissues generates disproportionate fragment sizes, ranging from single cells to fragments containing 100s to maybe 1000s of cells. This meant that the multiple domes that were set up may have contained vastly different amount of epithelial content. Starting with single cells, we are able to accurately control the initial starting cell number for each organoid culture in order to minimise variability across different experiments. We have now included this statement in the Methods section.

Single cells were used for the initial culture to enable equal cell numbers seeded into the Matrigel domes, which can be accurately quantified. Multiple domes (6-11) were set up for each genotype and treatment groups (control, HU or Estrogen) to enable sufficient material for downstream assays and passaging.

Page 13, lines 519-522.

- In line 135, is the 5-6 mice per condition of the age in weeks of the mice used?

We have now included the ages of mice used in this group.

Page 4, line 145.

- For the experiment performed for Figure 3, were there technical replicates or only biological replicates?

To obtain enough material for all experiments, all samples at all passages were set up in a minimum of 6-10 Matrigel domes of organoids. For all immunofluorescence experiments, 4-5 domes of organoids were pooled together to ensure robust number of organoids were counted.

We have now included this within the methods section.

Page 14, lines 572-573, updated Figure 3d-f.

- It would be helpful to show a representative example for the gating strategy used to derive the MFI data for Supplementary Figure 1A

The data was collected from the immunofluorescent imaging of tissue sections, and as such no gating strategy was involved. Tissue sections were imaged and analysed using Image J to calculate the MFI. We have now included a better description of this analysis in the methods section. Analysis for tissue sections: Tissues images were collected from a confocal using the tissue stitching tool to recreate the tissue section or from whole tissue images scanned on an AxioScan microscope. Using FIJI/ImageJ (version 1.53q), regions were drawn around all glandular/ductal structures, using DAPI as a counterstain. Mean fluorescent intensity (MFI) was normalised to the region area to account for differences in gland sizes and the multiple regions were averaged for each tissue sample. Page 15 lines 605-609.

- In figure 4B, it would be visually helpful to group cell types together, for example, not splitting up the luminal populations with the two basal populations in the middle

We have now changed the ordering of the figure to group the luminal cell types together. Ordering is now consistent as with other figures i.e. 4C/4E.

- Citations should be included to justify the use of certain markers for various populations in scRNAseq (paragraph starting on line 203)

We have cited the papers by Bach et al., [doi: 10.1038/s41467-017-02001-5] and Li et al., [doi: 10.1016/J.CELREP.2020.108566].

Page 6, line 249.

- The origin of the gene lists used for the scores derived in Figure 3C need to be shared

We used the gene list generated in the Saeki publication [doi: 10.1038/s42003-021-02201-2], reference 32. This is available as supplemental table 4 of their publication.

- For figure 4E, please list the control conditions before the HU conditions

We have now rearranged the figure by placing controls before HU conditions.

- AZD1480 should be clarified as a JAKi in the manuscript, rather than just assumed based on the figure label

We have now clarified the abbreviation.

Page 8, line 315-316.

- When it is stated on line 271 that cluster 4 is “defined by” high expression of Tspan8 and Thrsp, how was this determined? What was the log₂fc value for this gene in cluster 4 relative to all other clusters (via FindAllMarkers function)? Is this gene a top gene for the cluster or just one that was used for the DotPlot shown? These strategies were described clearly in the methods but could be stated in reference to this particular gene in the manuscript

We determined this via the FindAllMarkers function in Seurat to identify DEGs in the Brca2^{mut/WT} HU treated Passage 4 cells compared to the wildtype control, wildtype HU treated and Brca2^{mut/WT} control cells. This is included as Supplemental Dataset 1 in the manuscript. We then compared the

expression of these genes in the list via the dotplot (Fig 4f) and examined the list of DEGs for genes strongly associated with the mammary gland, which Tspan8 and Thrsp came within the top 30 genes. This is described in the results section page 7 lines 274-277. We have included the reference to the supplemental dataset for the readers. Page 7, line 277. The other genes in the top 30 list were either not strongly associated with the mammary gland, or a lower differential expression (*Aqp5*). We have softened the tone of this sentence, to state that cluster 4 contains high expression of mammary genes such as Tspan8, and since Tspan8, a cell surface marker, is expressed, it could be used as a surrogate to identify this cluster. This reduces the emphasis that this cluster is essentially defined by Tspan8. Page 8 lines 323-327.

- In figure 5, it would be helpful to show a proportion of cells graph as in Figure 4E to better appreciate the relationship of cluster 4 to the Brca2 genotype and the HU treatment

As we were only reporting on the passage 4 data in Figure 5, we had completed this analysis and displayed it as a stacked bar plot in Supplemental Figure a. As figures have been updated with new data, this is now in Supplemental New Figure 6a. It can be observed that cluster 4 (light teal colour) is expanded in the Brca2^{mut/WT} HU passage 4 cells compared to the other conditions.

- As all experiments are completed using mouse cells, the title should be edited such that Brca2 is written not in all caps

Thank you. We have now corrected this oversight.

- What is the meaning of neutrophil genes displayed as upregulated in epithelial cells (Supplementary 5B, D)? Might this relate to exposure to reactive oxygen species?

The genes upregulated in Brca2^{mut/WT} HU P4 clusters 1 and 3 relating to neutrophil showed to include genes related to gene involved in secreted proteins with antibacterial, antifungal and antiviral activity and in lactation. There does not seem to be any indication that these genes elevated in these pathways are involved in response to reactive oxygen species. For example, NRF2 signatures are not altered between conditions, as might be expected if ROS were induced. We have included some of the genes in the results section, and also referenced the supp dataset tables. Page 8 lines 295-308.

Reviewer #3 (Remarks to the Author):

In this manuscript, Najafabadi and colleagues explore the phenotype of BRCA2 heterozygous organoids expressing a pathogenic mutation and address the question of whether BRCA2 mutation per se affects the fate of mammary epithelial cell subpopulations using murine organoid models. Using acute versus chronic exposure to replication stress induced by HU and analyzing subpopulations by sc-RNAseq the authors find that Brca2mut/wt cells activate a transcriptional response to chronic replication stress that favors the outgrowth of hormone receptor-negative luminal cells. Moreover, they identify genes of the interferon response and mammary alveolar genes as responsible for this phenotype.

This is a very difficult an important question to address; indeed, BRCA2 mutated tumors are very different from BRCA1 mutated ones in that BRCA2 mutated tumors are generally luminal-like and HR+. This is very different from BRCA1-mutated tumors which are generally basal-like and triple-negative. The reason for this is still ill-defined.

Intriguingly, the authors find expanded mammary luminal populations that are HR- in Brca2mut/wt organoids just like what it has been found in BRCA1-mutated tumors; however, they do not comment on why or how do they explain that HR-luminal cells expand in their model whereas

BRCA2-mutated tumors are generally HR+. So then, what is the relationship between BRCA2 tumorigenesis in mice versus in humans? This manuscript falls short on answering this underlying question and needs to be addressed to determine whether their model recapitulates or not what happens in human BRCA2-mutated tumors and overall, what are the clinical implications of this work.

We agree with the Reviewer that the mechanisms accounting for the differences between BRCA1 vs BRCA2-mutant breast cancers are ill-defined, but also agree that this problem is a very difficult one to address in experimental systems. We make two points in response, although we cannot fully explain why HR- luminal cells in our Brca2 het organoids expand after HU challenge. First, while the Reviewer is correct to say that the majority of BRCA2 mutant cancers are HR+, consistent with cancers in the general population, a significant minority of BRCA2 mutant breast cancers are of the triple negative subtype. Our experimental model may represent the evolution of this less-frequent subgroup. Moreover, normal HR+ luminal cells are extremely difficult to expand in *in vitro* models (i.e doi: 10.1038/ncomms9786). Although the culture media we used (by StemCell Technologies) is one of the first that is reported to maintain HR+ expression during organoid culture, as documented by immunofluorescent staining and scRNA seq (Fig 3d and 4), it may nevertheless be insufficient to enable their outgrowth following HU challenge. These points are now summarized in revisions to the text of our paper.

Page 11 lines 419-421; 438-446.

We respectfully but firmly disagree with the Reviewer's view our manuscript "*needs to be addressed to determine whether their model recapitulates or not what happens in human BRCA2-mutated tumors and overall, what are the clinical implications of this work.*" Both Reviewers 1 and 2 acknowledge the interest, novelty and potential significance of our work. While the organoid model we have used is certainly relevant and widely deployed, it cannot fully recapitulate the complex events leading to human BRCA2-mutated tumours, or have direct clinical implications, and we believe it would be unreasonable to set such a high bar for publication.

Also, I wondered why they chose breast tissue from aged versus young given that BRCA2-mutated breast cancer occurs at early onset as compared to the general population.

We explain our choice as follows (from our response to Reviewer 1): "Adult mammary glands in mice are generally considered to form from 12+ weeks of age, which is equivalent to a young adult female, and mammary glands from an aged murine cohort are predicted to be equivalent to a mid-aged human female [doi: <https://doi.org/10.1016/B978-012369454-6/50074-1>]. We utilised young and aged mammary glands to eliminate possible epithelial differentiation changes. Since we confirmed that epithelial proportions and cell types were unchanged, we decided to work on the aged cohort, as the mammary glands would have been exposed to more hormonal cycles than younger mice, again mimicking scenarios similar to the increase in hormone cycle exposure in female human breast. This point is now explained in the text."

Page 3 lines 112-118.

There are some other major points that have to be addressed to consider publication as well besides these general ones that are detailed below:

Fig. 1D. The authors state that Het-aged mice display an increase in HR-luminal markers. However, in the graph, we can see that the Het-young mice display an increase, not the old ones, where the expression of those markers is even reduced compared to Wt-young.

We thank the reviewer for bringing this to our attention. There has been a labelling error. In order to align the text in the same direction for this figure, we had not realised the whole label was rotated, instead of each individual label. This has now been corrected in the revised Figure, which now conforms to the statement in the text.

Fig. S1B. In this figure, the gene expression does not correlate with the one shown in the heat map (fig 1D): in fact in S1B, the Het-aged mice display an increased expression of the genes CD14 and CD36. However, when looking at the expression of the same genes in fig. 1D we can see a strong expression (intense red box) in the Het-young mice whereas the Het-aged show a reduced level (white box). Could it be an error in the legend in Fig. 1D?

See comment above.

Fig.2D: how come HU-treated cells (both WT and Brca2-mutated) proliferate more than their not-treated counterpart? HU is known to slow replication so we expect the opposite trend. Moreover, the levels of BrdU+ cells differ from the ones in Fig5E under the same conditions.

Concerning Figure 2D, there may be a misunderstanding. An increased percentage of HU-treated cells exhibit BrdU incorporation during S phase, due to stalled or slow replication, but this is not equivalent to more proliferation. Please note that organoids were exposed to BrdU for 12 h, and so every cell entering but not exiting S phase during this period would be BrdU+ in our analyses.

Concerning Figure 5E, we agree that there are fewer BrdU+ cells after HU exposure in passages 2 or 4 (P2, P4) than in passage 0 (P0) shown in Figure 2D. We note that P0 organoid cultures were initiated using single cells, whereas P2 and P4 cultures were derived from passaging organoid fragments containing large numbers of cells, which may explain this difference in the percentage of cells entering S phase under different conditions.

Page 5 lines 174-182, Supplemental Figure 3b-c.

Fig. 3B: These results seem not conclusive as the error bars are quite big between -HU + HU in the mut/wt cells. As this is an important result, a complementary assay such as the FACS showing the number of BrdU+ cells in the two conditions as shown in Fig. 2 would be more convincing.

We have incorporated the new data, with more samples and the updated results [Fig 3b] remain consistent. Organoid numbers diminished in the WT HU treated group compared to WT control, and a slight, but not statically significant, decrease of organoid numbers in the Brca2^{mut/WT} HU treated group compared to Brca2^{mut/WT} control. We have now measured BrdU+ cells in the subsequent passages (P2 and P4), with no differences between the genotypes or treatments in these passages. This is now included in [new supp fig 3]. This shows that the dosage of HU utilised in this study is below the sublethal dosage, thus allowing us to investigate the impact of replication stress in mammary gland expansion across the two genotypes.

Figure 3, and Supplemental Figure 3b-c

Also, are these cells HR-luminal? Please specify in the legend.

These are whole organoids. Organoid domes were live imaged in order to quantify organoid numbers before being collected for downstream analysis. We have included this in the legend.

Fig 3D, S3D and S3E: The authors state that in figure S3D the multi-lineage organoids – corresponding to the black box, were present in 60, 33, 33 and 50 of Wt, Wt-Hu, B2 mut and B2 mut+HU; these numbers do not seem to correspond with the bar graphs. Please correct.

We have performed additional (n=3) samples and incorporated the new data. We have now updated all of the percentages accordingly, Figs 3D, new fig 4d-e.

Page 6 lines 210-220, Figure 3d-f and Supplemental Figure 4d-e.

Fig. 3F: There is quite some variability between different organoid cultures in Brca2 mut/wt which leads to a difficult assessment of the changes. It would be much more informative/convincing to show the paired organoids (HU-treated vs untreated) to determine whether the trend is similar in all or not and be able to conclude.

We have now included more samples (n=5) and updated the figure to show the paired data. Figure 3f.

Fig. 5G: the graph shows that both WT cells and Brca2mut/wt HU-treated show an increase in Tspan8+ cells when treated with IFN β compared to untreated cells however there is no comment on that in the text.

The differences in Tspan8 expression originally observed were small, and so we have updated the figure to include additional (n=5) samples. The updated figure shows a slight, but not significant increase of Tspan8 expression. We have now amended the text to reflect the updated figure.

Page 8, lines 323-329, Figure 5g.

What is the significant relevance of this if the difference is already there in the non-HU WT condition?

As noted above, in our revised analyses including more samples, there is no statistical difference in Tspan8 expression following IFN β or Jaki exposure. We have now amended the text to reflect the data.

Page 8 lines 323-329.

In general, this Fig. and the interpretation of the data are unclear. For example, Jaki is supposed to affect organoid size independent of proliferation but later on, is stated that it affects proliferation. Please reanalyze the data and clarify the conclusions. Then change accordingly the statements in lines 337-339.

We have now included additional (n=5) samples, updated the analysis. We assessed whether cells were able to recover from these treatments, and seeded single cells into new organoid cultures. Thus, treatments with Jaki did reduce organoid size (Figure 5f), once Jaki was removed cells were able to recover and form comparable organoid-formation capacity to controls (Figure 5h). We have included these statements in the text.

Page 8, lines 329-340, Figure 5e-i.

Fig. 6C. In the text they refer to this panel stating that Tspan8 expanded in the HR-luminal cells but there are no stats on the graph (or they are not significant) so please clarify this statement/graph.

We have performed additional (n=5) samples and have updated the figure, performed statistical analysis. We can confirm that Tspan8+ subpopulation was significantly increase in the Brca2^{mut/WT} HU treated group compared to the Brca2^{mut/WT} control, and there is no change in Tspan8+ cell proportion in the wildtype group. We have now updated the text to reflect our observations.

Page 9 lines 349-356, Figure 6c.

Fig. S5.A: In the WT-C organoids there is a similar proportion of cells belonging to cluster 4, similar to B2-HU. Moreover, following HU, in the WT there is an expansion of cluster 5, is it also associated with interferon response?

We have performed analysis on the WT-C cluster 4 population and identified upregulated of neutrophil mediated immune reponse pathways (New Supp Fig 6j). Whilst still immune related pathways, these pathways are not involved in interferon response, and so differ from the *Brca2*^{mut/WT} HU treated data observed in Fig 5c. We next performed the same GO term analysis with the WT HU treated cluster 5 population. However, when we performed the FindMarkers command to find DEGs in the WT HU vs the other 3 groups, the analysis returned 1 significantly DEG gene (*Clca3a2*). This is most likely due to the similar proportion of cells within this cluster in the WT HU in the P4 and the *Brca2*^{mut/WT} C in P4 conditions. As such there is not enough DEGs to perform this analysis. We then only looked at the WT samples and performed the analysis comparing control and HU treated wildtype organoids in passage 4. We observed DNA damage signalling pathways altered in this cluster (New Supp Fig 6h, and supplemental datasets 3).
Page 8 lines 295-305, Supplemental Figure 6g-j.

Fig. S5.B: cluster 1 of *Brca2*-HU treated cells is associated with IFN response, how about cluster 1 of WT-C, which is even more present than in B2-HU?

The analysis of WT_P4 vs rest for cluster 1 showed similar neutrophil terms (New Supp Fig 6i-j).
Page 8 lines 305-308 Supplemental Figure 6g-j.

Minor

Introduction: When referring to Ref. 6 the authors state that there are no phenotypic changes in myoepithelial cells in *Brca2* mutated tumors whereas the ref. states the opposite. Please correct.

The manuscript by Ding et al., [doi: 10.1038/s41467-019-12125-5] on myoepithelial cells is complex, and the reviewer is correct that our statement of no differences in the *BRCA2*-mutant basal cells is incorrect. In our original draft, our statement was in reference to the fraction of p63+ TCF7+ BA cells in *BRCA2* carriers, but this was not specified appropriately, since Ding et al., go on to show some differences, for example, in gene expression of *BRCA2*-mutant basal cells. As the specifics of this report are not particularly germane to the overall content of our manuscript, we have amended the text to state: "... myoepithelial cells from *BRCA2* carriers have relatively minor molecular differences from wildtype cells compared to *BRCA1* carriers..." We believe this statement more accurately reflects the work of Ding and colleagues—who do report more substantive differences in B1 carrier basal cells—without detracting from the greater narrative flow of the Introduction. We thank the reviewer for raising this concern.

Page 2 lines 52-53.

Fig. 3B. Het control should be labeled in light blue to follow the same color logic as in the other graphs and avoid confusion.

We thank the reviewer for noticing this discrepancy. We have updated the colour of the labels to make this consistent.

Fig. 3E. The authors state that there is a significant expansion in the HR- compartment although in the figure we can only see that the Het-HU organoids display an increase in the CD49+ but no in the DN compartment, where there is no difference across the conditions.

We have clarified this to refer to the CD49b+ HR- luminal subpopulation.

Fig 4D: in the text, the authors state that the expression of the luminal alveolar gene Aqp5 is elevated in the B2_HU_P4, however, when looking at the figure we can see that for this gene there is no difference in expression when compared to WT P0 or P4.

The analysis for this figure utilised the FindMarkers feature to examine the Brca2^{mut/WT} HU treated cells compared with the DEGs in the other 3 groups (Brca2^{mut/WT} control, Wildtype control and wildtype HU treated). We performed this analysis on the passage 4 cells. We did not do a comparison between passage 0 and 4. The dot plot in Figure 4f displays the expression data from the genes in all groups and conditions. It is true that there is minimal expression difference between Brca2^{mut/WT} HU treated passage 4 and some of the passage 0 groups, but when examining the passage 4 'combined expression' of Brca2^{mut/WT} control, Wildtype control and wildtype HU treated groups the analysis showed a difference in expression. The analysis data including logFCs was included as the supplemental dataset 2.

Fig. 5F: The interpretation that Jaki reduced organoid size only in the HU-treated Brca2- mutated organoids seem to be an overestimation considering the variability in the replicas in this condition; is it not significant the reduction observed in the other groups upon AZD1480 treatment?

We have now included additional (n=5) samples, We now observe a consistent statistical reduction in all groups following Jaki treatment. We have updated the analysis and clarified the results and conclusions accordingly. Figure 5e-i

Fig. S3A: The authors conclude from this figure that all the epithelial sub-populations display similar levels of DNA damage across the different groups; however, they fail to comment on why there are different levels of damage in the different lineages, in particular why the treated WT accumulate more gH2AX in the basal and HR+ compartment.

Mammary epithelial subpopulations have differences in many cellular functions. Studies have shown differences in telomere lengths between the different lineages [doi.org/10.1016/j.stemcr.2013.04.003], and in cell division kinetics [doi.org/10.1038/ncomms9487]. A BioRxiv manuscript also suggests that functional capacity of HR repair differs across the mammary lineages [doi.org/10.1101/2021.05.14.444217]. We have included a statement addressing this point. We extended the ANOVA analysis to compare between WT and Brca2^{mut/WT} groups, and although there may be a trend towards a visual increase in DNA damage in the WT basal and HR+ luminal populations, this was not significant. We have updated the figures to include the statistical data. Page 5 lines 171-174, Supplemental Figure 3a.

REVIEWER COMMENTS

Reviewer #1 (Remarks to the Author):

The revised manuscript with new experiments and discussions has sufficiently addressed the previous comments.

Reviewer #2 (Remarks to the Author):

The authors addressed some of the concerns raised by the reviewers, however issues with the data analysis, presentation, and data interpretation still persist.

- 1) IF images have to display the staining so the reader can appreciate the data (only some have this).
- 2) Flow cytometry and CyTOF analysis only show data summary - Flow plots, gating strategy for sorting must be displayed.
- 3) CYTOF data should display data points for all samples (n=3 according to text).
- 4) For CYTOF data it is unclear to why samples are organized as shown (wt aged, het young, het aged, wt young) - is this a clustering organization? if not, please pair each sample with its appropriate control for better data interpretation.
- 5) panel 1E - why these markers were utilized? cd14 can mark myeloid cells - where is the data quantification? if not significant, does that mean that the CYTOF did not work?
- 6) Flow plots showing the flow strategy for the isolation of cells for organoids must be included, and as well the gates utilized. On 1G - it is not clear whether organoids were derived from the same animal, or from 3 independent animals. This is important given that there is no mentioning of whether animals were analyzed with the same estrous cycle
- 7) Fig.2 measures what seems to be 2 populations of HR- luminal cells, while 1G only reports one - what are the differences here, and why both populations were not presented on 1G?
- 8) Wholemount staining confirmed the presence of both single lineage and multilineage organoids (Sup Fig 1h), confirming their stem cell-like capacity - This analysis is not adequate for the determination of stem-like properties. Lineage transcription, transplantation, serial passage needs to be employed to support the authors conclusions.
- 9) there is still a lack of understanding to why HR negative cells are expanded in Brca2 wt/mut organoid cultures, why this was not observed in mammary tissue
- 10) 2B should be moved to sup.
- 11) the BrDU analysis needs to be carefully interpreted. For example, in basal cells HU induces the same increase comparing untreated and treated het, then observed when comparing wt untreated to wt treated. Again, it is unclear the flow strategy to separate all of these cell types.
- 12) across the manuscript, the authors refer to brca2 deficient cells as het or Brca2mt/wt , or wt versus wildtype - please pick one and use it consistently across the manuscript
- 13) analysis of cell populations varies from flow measurement to image analysis - the authors should

show that both approaches yields the same results in at least 1 experiment, so support that they can reliably utilize these strategies alone.

14) overall, the changes quantified across the manuscript are minimal, raising concerns about the reproducibility of the study and the biological relevance of the study

15) scRNAseq indicates cycling populations – are those equally distributed across the wt and het conditions?

16) scRNAseq indicates lineage score analysis – are those equally represented across the wt and het conditions?

17) on Fig.1 the authors showed that luminal HR- were expanded in het organoids – why this difference is not represented on scRNAseq data? Where are the DN or C49b+ cells on scRNAseq? Also, how many replicates per time point?

18) are the differences shown on 3F statistically significant?

19) the differences across treatments on Fig.5 are marginal, probably because the influence of the selected targets is acting in one cell population specifically, rather than across all cells/organoids – so maybe focusing these analyses on HR- luminal cells would show real differences

20) what are the levels of tspan8 on other cell populations? scRNAseq shows that other cells also express it.

21) to define a specific role of tspan8 on hr- luminal cells, the authors should have mixed Crispr targeted hr- luminal cells with basal and hr+ untargeted cells.

Reviewer #3 (Remarks to the Author):

The authors have satisfactorily revised the manuscript.

Just to clarify to the reader, in the discussion (line 479 onwards) it should be clearly stated that these findings are based on a mouse organoid model and that the expansion of luminal population may contribute to explain the ~20% of Brca2 tumors that are HR-.

We thank the Editors for the opportunity address the remaining points raised by Reviewer #2, which largely request us to revise how we display certain datasets. We have also responded to the single minor point raised by Reviewer #3. Our responses are summarized below, and the manuscript has been revised accordingly.

Reviewer #2 (Remarks to the Author):

The authors addressed some of the concerns raised by the reviewers, however issues with the data analysis, presentation, and data interpretation still persist.

1) IF images have to display the staining so the reader can appreciate the data (only some have this).

We have now included representative IF images alongside the analyses for all of the figures.

2) Flow cytometry and CyTOF analysis only show data summary - Flow plots, gating strategy for sorting must be displayed.

For flow cytometry: Plots and gating strategy (for primary cell sorting, organoid culture, gH2AX, BrdU and Tspan8 staining and Tspan8 CRISPR experiments) have been added to Supplemental Figures 14-18 and in the Materials and Methods section.

For CyTOF data: CyTOF FACS plots from the initial QC (done in FlowJo) and UMAPs from the QC (done with Scanpy) have been added to Supplemental Figures 11-12 and Methods.

3) CYTOF data should display data points for all samples (n=3 according to text).

The CyTOF data as presented in Figure 1D show the mean of the three samples per group. We have explained this in the Figure Legends for Figure 1. Given that no difference in marker expression was observed, showing each individual data point would make the figure unclear and difficult to visualize, and would not, in our opinion, provide any important additional information. We emphasize that all of the processed and unprocessed CyTOF data have been deposited in Mendeley Data, so that interested readers will be able to probe the data in greater detail should they wish.

4) For CYTOF data it is unclear to why samples are organized as shown (wt aged, het young, het aged, wt young) - is this a clustering organization? if not, please pair each sample with its appropriate control for better data interpretation.

The original order of the groups presented in Figure 1D was determined using a clustering approach. However, we agree with the Reviewer that as presented, the ordering of the groups did not provide high visual clarity for readers. We have thus re-generated the heatmap to show the groups in a more logical order (Wildtype Young, Brca2^{mut/WT} Young, Wildtype Aged, Brca2^{mut/WT} Aged). Note that, per our analyses and the text, no marker was significantly differentially expressed between the wildtype and Brca2^{mut/WT} groups in either age group.

5) panel 1E - why these markers were utilized? cd14 can mark myeloid cells - where is the data quantification? if not significant, does that mean that the CYTOF did not work?

We had noted in the results (page 3 lines 108-112) that although the CyTOF data was not statistically significant, several HR- luminal markers including E-Cadherin and CD14 displayed slightly increased expression in the Brca2^{mut/WT} aged HR- luminal populations compared to wildtype counterparts (Fig 1d). E-Cadherin and CD14 displayed equivalent protein expression between the Brca2^{mut/WT} glands compared to wildtype further confirmed no statistical differences validating the CyTOF data (Fig 1e,

Supp Fig 1b-c). Data quantification is provided in Supplemental Figure 1b-c. To ensure only epithelial material was quantified, regions were drawn around all glandular/ductal structures. The method of quantification was included in the materials and methods sections [page 15, lines 604-608].

Overall, the tissue and CyTOF staining showed equivalent results indicating that both techniques had worked.

6) Flow plots showing the flow strategy for the isolation of cells for organoids must be included, and as well the gates utilized. On 1G - it is not clear whether organoids were derived from the same animal, or from 3 independent animals. This is important given that there is no mentioning of whether animals were analyzed with the same estrous cycle

We have now included the flow strategies I Supplemental Figures 13-14. Figure 1F is a representative image from organoids derived from one animal. The figure legend was updated to make this clearer. Figure 1G is quantification in organoids of the different epithelial populations from four independent mice. The figure legend has been updated to make these points clear.

7) Fig.2 measures what seems to be 2 populations of HR- luminal cells, while 1G only reports one – what are the differences here, and why both populations were not presented on 1G?

The experiments in Figure 1 use the primary, non-cultured material. All subsequent figures and supplemental data are derived from organoid cultures. The flow plots from the luminal population showed a range of expression based on CD49b (Supplemental figure 14). Using single stained and fluorescence “minus 1” controls (where a control tube of cells is stained with all fluorochromes used in the experiment except one), we divided the CD49b population into 2 (CD49b+ cells and double negative (DN)). CD49b+ and DN populations did not express Sca1 and subsequently grouped within the HR- Luminal population.

8) Wholmount staining confirmed the presence of both single lineage and multilineage organoids (Sup Fig 1h), confirming their stem cell-like capacity – This analysis is no adequate for the determination of stem-like properties. Lineage transcription, transplantation, serial passage needs to be employed to support the authors conclusions.

We agree with the reviewer that we did not test for the full suite of stem cell properties necessary to justify our original phrasing. As this point is not central to our major conclusions, we have amended the phrasing to draw attention only to the capacity of BA and HR- luminal cells to generate multiple epithelial lineages *in vitro*, consistent with other reports as cited in the manuscript. We believe this change better reflects our results.

Page 4 line 136

9) there is still a lack of understanding to why HR negative cells are expanded in Brca2 wt/mut organoid cultures, why this was not observed in mammary tissue

We note that HR negative cell expansion has recently been reported in non-malignant epithelium derived from the breast tissues of *BRCA2* mutation carriers (Karaayvaz et al 2020. DOI: 10.1126/sciadv.aay2611), consistent with our work. We have revised the text to note this point. We also emphasize here that the expansion of HR- luminal cells we observe in organoid cultures happens specifically in the context of HU treatment. Human females are exposed to environmental genotoxic stresses over the course of decades, whereas the primary murine tissues utilized herein were derived from animals <1 year of age raised in carefully controlled environments. This

distinction motivated the HU treatment experiments (mimicking stressors experience by humans over time) in the first place (as note in the Results; lines 164-165).

Page 11, lines 445-446.

The Reviewer is correct to note that there is still a lack of understanding concerning why different epithelial cell populations behave differently in organoid cultures versus mammary tissue. Although in our work we have used widely accepted methods for *in vitro* mammary organoid cultures of mouse/human breast epithelium, it is widely accepted in the field that culturing HR+ luminal cells is a challenge; sustaining the outgrowth of hormone-receptor positive mouse mammary cells long term has only been reported in a single organoid protocol (doi.org/10.1038/ncomms13207). However, to address such issues would go far beyond the reasonable scope of our paper.

10) 2B should be moved to sup.

We are willing to comply with the Reviewer's request, but would prefer not to do so for the following reason. Fig 2B shows important data demonstrating that DNA damage marked by gH2AX staining occurs after HU exposure in all cell populations and genotypes. We therefore believe that retaining this Figure in the main text rather than Supplementary Information would make our scientific narrative clearer, and the results simpler to interpret.

11) the BrdU analysis needs to be carefully interpreted. For example, in basal cells HU induces the same increase comparing untreated and treated het, then observed when comparing wt untreated to wt treated. Again, it is unclear the flow strategy to separate all of these cell types.

The Reviewer is correct that HU induces a similar increase in BrdU+ staining amongst basal cells from both the WT and Brca2^{mut/WT} genotypes, but this increase in both cases is not statistically significant (WT untreated/ HU treated, p=0.163; Brca2^{mut/WT} untreated/ HU treated, p=0.795). There is, however, a statistically significant difference between HU-treated WT vs Brca2^{mut/WT} samples, as noted in the figure. A similar point was raised by Reviewer #1 (and 3) in the original manuscript, and so we have carefully interpreted the BrdU analysis in the revised version to clarify this point. To clarify the flow strategy, we have included the flow gating in Supplemental Figure 17. BrdU positive cells can be clearly identified in all of the epithelial subpopulations.

12) across the manuscript, the authors refer to brca2 deficient cells as het or Brca2mt/wt , or wt versus wildtype – please pick one and use it consistently across the manuscript

Thank you for noting this point - we have revised the text, figures and figure legends to be consistent.

13) analysis of cell populations varies from flow measurement to image analysis – the authors should show that both approaches yields the same results in at least 1 experiment, so support that they can reliably utilize these strategies alone.

Our results clearly show that expansion of the HR- luminal population after HU exposure is evident in both our wholemount image analysis and flow measurements. Thus, both methods support our conclusion. However, there are underlying technical differences between these methods which render them quantitatively distinct. For e.g., flow measurements can separately quantitate HR+ and HR- luminal cells, whereas wholemount staining cannot. Thus, it would not be possible to obtain exactly the same results owing to the fundamentally distinct nature of quantification in the two assays. Additionally, we refer in the methods sections that wholemount staining was performed on

a pool of 4-5 Matrigel domes and 1-2 Matrigel domes were used for flow cytometry analysis. For tissue glands 1-2 glands were used for each technique, thus it would not be possible to use the same glands or organoids for the wholemount and flow analyses.

14) overall, the changes quantified across the manuscript are minimal, raising concerns about the reproducibility of the study and the biological relevance of the study

We believe that our results are robust, and demonstrate their reproducibility, with many experiments performed on >5 independent biological samples, using several different technical approaches to substantiate our conclusions. The range of variability that we observe is consistent with expectation in these types of experiments, using primary tissues and organoid cultures. Our results address a problem of high interest and biological relevance. In addition, our work on non-transformed mammary epithelium organoids has separated itself from most of the other studies in the field, thus is highly relevant to tissue biology.

15) scRNAseq indicates cycling populations – are those equally distributed across the wt and het conditions?

Figure 4 D and E represents the proportion of Basal (Dark blue) and HR- luminal (dark green) cycling populations between the genotypes, passages and treatments. As shown in Figure 4D and quantified in 4E these cycling populations are not equally distributed across the conditions.

16) scRNAseq indicates lineage score analysis – are those equally represented across the wt and het conditions?

Figure 4C represents the pooled data for the lineage scores. The individual genotypes, passages and treatments of each lineage score was shown in Supplemental Figure 6a – Basal lineage, 6b – HR- luminal lineage and 6c – HR+ luminal lineage. As observed from the data, the lineage scores identifying the lineage subtypes were equally represented across all conditions.

17) on Fig.1 the authors showed that luminal HR- were expanded in het organoids – why this difference is not represented on scRNAseq data? Where are the DN or C49b+ cells on scRNAseq? Also, how many replicates per time point?

Our scRNAseq data (Figure 4E) does show that the HR- luminal population is expanded in the BRca2^{mut/WT} HU treated group. The flow plots from the luminal population showed a range of expression based on CD49b (Supplemental figure 14). The HR- luminal cell population, which contains both CD49b+ and DN cells, are represented by the light and dark green bars (Figure 4E). Concerning replicates, the Figure legend notes that we analysed a total of 34, 200 cells. We performed n=3 biological samples for each time point. This constituted a total of 24 samples (n=3 individual biological sample for each genotype (wildtype and Brca2^{mut/WT}), passages (P0 and P4) and treatment (Control and HU) of organoids), with an average of 1500 cells sequenced per sample and we have now explicated stated this in the figure legend.

18) are the differences shown on 3F statistically significant?

We thank the reviewer for pointing this out and we have added ns=non-significant in the figure legend.

19) the differences across treatments on Fig.5 are marginal, probably because the influence of the selected targets is acting in one cell population specifically, rather than across all cells/organoids – so maybe focusing these analyses on HR- luminal cells would show real differences

All data presented in Figures 5E, G and I using flow measurements do refer specifically to the HR- luminal population (as indicated in the respective figure legends). Thus, the overall interpretation of the data as reported in the text is in reference to the HR- luminal population.

20) what are the levels of Tspan8 on other cell populations? scRNAseq shows that other cells also express it.

We have included the flow plot strategy for Tspan8 to show expression in the other populations, and the proportion of Tspan8+ cells in the epithelial populations (Supplemental Figure 9b).

21) to define a specific role of Tspan8 on hr- luminal cells, the authors should have mixed Crispr targeted hr- luminal cells with basal and hr+ untargeted cells.

The experiment proposed by the Reviewer is technically challenging, would take many weeks, and might be subject to difficulties in interpretation, given that Tspan8 is reported to perform different biological functions in breast epithelial cell populations. We therefore believe that it falls beyond the reasonable scope of this paper. Instead, we have revised the text to say that our experiment does not rule out the possibility that the observed effect of Tspan8 depletion could arise from effects on basal or HR+ cells. However, their low frequency (Supplemental Figure 9B), and our clear demonstration that expansion of HR- cells after HU exposure is abrogated by Tspan8 depletion, make such an interpretation marginal.

Page 12, lines 474-475.

Reviewer #3 (Remarks to the Author):

Just to clarify to the reader, in the discussion (line 479 onwards) it should be clearly stated that these findings are based on a mouse organoid model and that the expansion of luminal population may contribute to explain the ~20% of Brca2 tumors that are HR-.

We have updated the discussion to indicate that these models were generated from murine tissues.

REVIEWERS' COMMENTS

Reviewer #2 (Remarks to the Author):

The authors have done a great job addressing all of my concerns.

REVIEWERS' COMMENTS

Reviewer #2 (Remarks to the Author):

The authors have done a great job addressing all of my concerns.

We thank the Editors and reviewers for their comments throughout this review process.